

# GRUN: An observations-based global gridded runoff dataset from 1902 to 2014

Gionata Ghiggi[1], Vincent Humphrey[1], Sonia I. Seneviratne[1], Lukas Gudmundsson[1]

[1] Institute for Atmospheric and Climate Science, ETH Zurich, Universitaetstrasse 16, 8092 Zurich, Switzerland

*Correspondence to*: Gionata Ghiggi (gionata.ghiggi@gmail.com)

**Abstract**

Freshwater resources are of high societal relevance and understanding their past variability is vital to water management in the context of current and future climatic change. This study introduces a global gridded monthly reconstruction of runoff covering the period from 1902 to 2014. In-situ streamflow observations are used to train a machine learning algorithm that
predicts monthly runoff rates based on antecedent precipitation and temperature from an atmospheric reanalysis. The accuracy of this reconstruction is assessed with cross-validation and compared with an independent set of discharge observations for large river basins. The presented dataset agrees on average better with the streamflow observations than an ensemble of 13 state-of-the art global hydrological model runoff simulations. We estimate a global long-term mean runoff of 37419 km$^3$ yr$^{-1}$ in agreement with previous assessments. The temporal coverage of the reconstruction offers an
unprecedented view on large-scale features of runoff variability also in regions with limited data coverage, making it an ideal candidate for large-scale hydro-climatic process studies, water resources assessments and for evaluating and refining existing hydrological models. The paper closes with example applications fostering the understanding of global freshwater dynamics, interannual variability, drought propagation and the response of runoff to atmospheric teleconnections. The GRUN dataset is available from the ETHZ Research Collection at https://doi.org/10.3929/ethz-b-000324386 (Ghiggi et al., 2019).

**1 Introduction**

Water is one of the most important natural resources for human development and its availability affects water supplies, agricultural yields, energy production as well as infrastructures safety and operation. Two-thirds of the global population are currently exposed to severe water scarcity (Vörösmarty et al., 2010; Kummu et al., 2016; Mekonnen and Hoekstra, 2016) and the most recent annual risk report of the World Economic Forum (WEF, 2018) lists the water crises as one of the largest
global risks in terms of potential impact and likelihood. While river flow is regularly used to assess regional renewable freshwater resources (Vörösmarty et al., 2000; Oki and Kanae, 2006; Veldkamp et al., 2017; Munia et al., 2018), there is to date no publicly available global dataset providing observation-based estimates of the evolution of runoff and river flow throughout the 20$^{th}$ and the early 21$^{st}$ century. In the last decades, several international initiatives promoted the launch of modelling inter-comparison projects with the aim to improve the representation of the terrestrial water cycle in global



hydrological models (Dirmeyer et al., 2006; Dirmeyer, 2011; Haddeland et al., 2011; Harding et al., 2011; Van Den Hurk et al., 2011; Warszawski et al., 2014; Van Den Hurk et al., 2016; Schellekens et al., 2017) as well as to develop tools to refine regional hydrological predictions in data sparse regions (Sivapalan, 2003; Blöschl et al., 2013; Hrachowitz et al., 2013). In the meantime, a widespread decline in the number of streamflow monitoring stations has also been reported (Shiklomanov et

al., 2002; Fekete and Vörösmarty, 2007; Fekete et al., 2012, 2015; Laudon et al., 2017) and alternative estimates of streamflow are thus needed for reconstructing past large-scale runoff variability, not only during the early century but also in the recent decade.

In this contribution, we use a recently published collection of in-situ streamflow data (Do et al., 2018; Gudmundsson et al., 2018b) in combination with a century-long reanalysis (Kim et al., 2017) to fill this gap. To this end, this study introduces a

global gridded reconstruction of monthly runoff covering the period 1902 to 2014 at a 0.5° spatial resolution. Runoff is here defined as the amount of water drained from a given land unit (i.e. grid cell) eventually entering the river system, including groundwater flow and snow melt. The methodology builds upon previous work where gridded runoff rates estimates were obtained for Europe (Gudmundsson and Seneviratne, 2015, 2016). Hereafter, these two papers are referred as GS15 and GS16 respectively. Monthly observations of precipitation, temperature and observed runoff rates from small catchments are

used as input for a machine learning (ML) algorithm to learn the runoff generation process without the explicit description of the involved hydrological processes. Gridded precipitation and temperature data are then used to predict runoff rates also in ungauged regions. The reconstruction accuracy is evaluated against runoff observations not used for model training and compared with the accuracy of an ensemble of global hydrological models simulations forced with the same precipitation and temperature data. The paper concludes with a section illustrating the potential of the newly established data product

(GRUN) for climatological, hydrological and environmental studies.

## 2 Data

### 2.1 Modelling Data

#### 2.1.1 Atmospheric forcing

Gridded observations of precipitation and temperature data are obtained from the Global Soil Wetness Project Phase 3

(GSWP3) dataset (version 1.05) (Kim et al., 2017). GSWP3 is a dynamically downscaled and bias corrected version of the 20th Century Reanalysis (20CR) (Compo et al., 2011). The dataset covers the period 1901 to 2014 and is available on a regular 0.5° x 0.5° grid at 3-hourly resolution. The sub daily data are aggregated to monthly means and bilinearly interpolated to a cylindrical equal area (CEA) grid composed of cells with an area of 2500 km$^2$ and a spatial resolution of approximately 50 km.

### 2.1.2 Runoff observations

Monthly runoff observations are derived from the Global Streamflow Indices and Metadata Archive (GSIM) (Do et al., 2018; Gudmundsson et al., 2018b). This dataset includes a collection of 35'002 streamflow stations obtained by merging existing international and national databases. GSIM provides a wide range of time series indices at monthly, seasonal and

yearly resolution. Here timeseries of monthly mean streamflow are considered. The data selection and pre-processing of these observations is detailed in Sect. 3.1.

## 2.2 Validation Data

### 2.2.1 Observed continental-scale river discharge

Observed monthly river discharge from 718 large river basins is taken from the Global Runoff Data Centre (GRDC)

Reference Dataset (https://www.bafg.de/GRDC/EN/04_spcldtbss/43_GRfN/refDataset_node.html). The dataset contains a selection of streamflow stations with a basin area greater than 10'000 km² and corresponding catchment shapefiles. The temporal coverage of each time series is longer than 20 years. These timeseries are removed from GSIM to be sure that an independent observational set is used for model evaluation (see Sect. 3.3).

### 2.2.2 Global hydrological models' simulations

The Inter-Sectoral Impact Model Intercomparison Project (ISIMIP) offers a framework to compare simulations and to quantify the uncertainty across hydrological and land surface models forced with equal inputs (Warszawski et al., 2014). The accuracy of GRUN is benchmarked against runoff simulations for the period 1971-2010 from 13 state-of-the-art global hydrological models (GHMs) participating in the second phase of ISIMIP2a Water (Gosling et al., 2017). The considered GHMs simulations are driven with the GSWP3 forcing and do not account for human impacts on river flow (nosoc scenario).

Further detail on the ISIMIP2a simulation setup can be found at https://www.isimip.org/protocol/#isimip2a.

## 3 Data selection and pre-processing

### 3.1 GSIM time series selection and pre-processing

**Step 1. Sub-setting GSIM stations and conversion of flow volumes to runoff rates**

Runoff is defined here as all the water draining from a small land area. Runoff cannot be observed directly, but at monthly

time scales, total catchment runoff can be assumed to equal the sum of streamflow if storage during water routing (e.g. dams, reservoirs) and river water losses (through e.g. channel and lake evaporation, irrigation) are negligible. If these assumptions are met, runoff rates can be obtained by dividing the GSIM river discharge with the station's upstream catchment area. Successively, for catchments which area is comparable to the size of the grid cells of the atmospheric forcing data, it is possible to use catchment runoff to derive an observational estimate of the runoff rate in the corresponding grid-cells.



To retrieve accurate estimates of grid-cell runoff, only GSIM stations fulfilling the following criteria have been selected for further analysis:

1. The timeseries has observations within the period 1902-2014 when GSWP3 forcing is available.

2. The original data provider reports an estimate of the drainage area. This choice is made to have the possibility to verify the geographic location of the station as well as to assess the reliability of the automated delineation of the drainage area using a digital elevation model as provided in GSIM (Do et al., 2018).

3. GSIM provides the shape of the drainage area and the quality of the catchment delineation is flagged as "*medium*" or "*high*". This criterion imposes that the difference between the drainage area reported by the data provider and the one estimated by GSIM differ less than the 10 % (Do et al., 2018).

4. The drainage area is between 10 and 2500 km$^2$. Very small catchments ($< 10$ km$^2$) are discarded because the uncertainty in the drainage area can significantly affect the magnitude of the runoff rates. On the other hand, catchments larger than 2500 km$^2$ are removed because their drainage area spans too many grid-cells of the atmospheric forcing.

Based in these criteria, 10042 GSIM stations are selected for further analysis.

**Step 2. Correction for mislabeled missing values**.

Manual investigation of monthly runoff time series revealed the occurrence of multiple consecutive months with runoff rates exactly equal to 0 mm/day, in disagreement with the observed regional runoff pattern and the climatological runoff signature. These artefacts likely stem from a misleading treatment of missing measurements (e.g. due to damaged sensors). To identify such likely missing values, all timeseries are screened for the presence of more than 3 consecutive months with values of zero. If this pattern occurs, all zero values in the monthly timeseries are set to "missing".

**Step 3. Remove timeseries with unrealistic runoff rates and short temporal coverage**

The following criteria have been adopted to remove observations which are physically very unlikely:

1. Remove timeseries with only missing values

2. Remove timeseries with negative monthly runoff rates

3. Remove timeseries with less than two years of observations

4. Remove timeseries with monthly runoff rates higher than 2000 mm/month

This screening step gives a selection of 8510 stations.

**Step 4. Homogeneity testing**

River discharge time series can show temporal changes in the hydrological signature because of changing instrumentation, recalibration of rating curves, flow regulation (i.e. dam construction) and other human activities (i.e. irrigation). Automated identification of such breakpoints is usually done using statistical tests (Gudmundsson et al., 2018b). GSIM used a general-purpose procedure that was applied to all indices/time scales. In this study, the following two target-oriented change point detection methods are applied after log-transformation of the time series:

1. Univariate normal change point in variance (Chen and Gupta, 2012).



2.  Univariate normal change point in normal mean and variance (Chen and Gupta, 2012).

Runoff timeseries are discarded when any of the two change point tests is significant at the $p<0.01$ level.

Figure 1 shows three river flow timeseries with different types of detected change points. Figure 1a illustrate the ability of the tests to identify changes in low flow regulation or low flows measurement precision. Figure 1b display the detection of

sudden drops in the time series, e.g. caused by dam construction, river diversion or measurement errors, while Fig. 1c shows the potential in spotting subtle changes in river discharge variability possibly induced by reservoir operations.

The homogeneity testing procedure resulted in a final selection of 7627 stations which are likely not impacted by flow regulation.

### 3.2 Assigning GSIM time series to grid cells corresponding to the atmospheric forcing data

To give equal importance to high-latitude and tropical observations, the entire modelling procedure is conducted on cylindrical equal area (CEA) grid composed of cells with an area of 2500 km$^2$ and a spatial resolution of approximately 50 km. The final gridded runoff product is however projected back onto the WGS84 grid of the atmospheric forcing data.

Because of the high density of stations in some regions and the typically elongated shape of the drainage area, many runoff observations span multiple cells of the CEA grid. Thus, an observational runoff time series representative of each cell is

retrieved as follow:

1.  Project the GSIM catchment shape to CEA
2.  For each grid cell:
    a.  Select those catchments of which the drainage area intersects the grid-cell.
    b.  At each time step, take the median runoff rate of the selected catchments.

Besides reducing the over-sampling in high station density regions, this step also smooths out some sub-grid variability. Additionally, it also can reduce the effect of potential outliers (i.e. stations that have exceptionally high or low runoff rates compared to their neighbors). To avoid inhomogeneities arising by the concatenation of different runoff timeseries, the observational runoff time series at each grid cell is submitted to another homogeneity testing run (as described in Sect. 3.1, Step 4).

The procedure resulted in 5544 grid cells usable for model training, covering 9.3% of the total land area and corresponding to 2799463 monthly runoff rate observations. Hereinafter, the grid-cell runoff timeseries are referred to as the runoff observations and Fig. 2 shows their spatio-temporal coverage.

### 3.3 Selection and pre-processing of GRDC time series

To obtain an independent dataset for assessing the accuracy of GRUN in large river basins, streamflow stations with catchment area larger than 50'000 km$^2$ are selected from the GRDC reference dataset. Although most of these stations are



included in the GSIM collection, they are not used for model training because only catchments with area smaller than 2500 km$^2$ are used to derive grid-cell runoff observations (Sect. 3.1, Step 1).

The GRDC time series are subject to the pre-processing steps 2 to 4 detailed in Sect. 3.1 to reduce the impact of inaccuracies and streamflow records heavily impacted by humans. Finally, only the most downstream stations of each river are selected. This procedure results in a selection of 214 large river basins.

## 4 Methods

### 4.1 Model Setup

For the first time, GS15 and GS16 have used a ML algorithm to estimate monthly runoff at continental-scale and Ghiggi (2018) explored the utility of a wide range of algorithms to improve the task. Based on these findings, the present study employs the Random Forest (RF) algorithm (Breiman, 2001). RF is a ML algorithm which averages a set of randomized regression trees (Breiman et al., 1984) trained on different subsets of the original data. A regression tree divides progressively the predictor space into high-dimensional rectangles by means of recursive binary splits. The predicted value of a new observation is the average of the observations used in the training process located in the region of the predictor space to which the new predictor values belongs. By averaging the predictions of several randomized regression trees build on different training data, RF improves the final accuracy of the runoff estimates.

The monthly runoff rate (R) is modelled as a function of monthly precipitation (*P*) and monthly near surface temperature (*T*) as

$$R_{s,t} = f\left(\tau(P_{s,t}), \tau(T_{s,t})\right) \quad (1)$$

where:

- f corresponds to the RF model (RFM);

- *s* represents the identifier of the CEA grid cell;

- *t* is the timestep;

- $\tau$ is a time lag operator that provides information about meteorological conditions of the past six months to allow the RFM to approximate water storage effects that influences the runoff generation process. This differs from GS15 and GS16 which used a time lag operator of 12 months. The reasons behind this change are a reduction in training time of RFM and to decrease collinearity between predictors (caused by the seasonal cycle).

Both precipitation and runoff observations are log-transformed before model training to adjust the skewed distribution of the data and avoid that only a small number of high flow events dominate the optimization of the squared error loss. Once the

RFM is trained, gridded precipitation and temperature data are fed to the model to obtain the final runoff reconstruction. Finally, the log-transformation of the predicted runoff values is inverted to derive runoff rates in conventional units.

## 4.2 The GRUN reconstruction

Accurate predictions of a machine learning algorithm are conditioned to training of the model with observations. The use of different training observations has the potential to generate different outcomes if the model is not able to generalize the relationship between the response (i.e. runoff) and the predictors (i.e. precipitation and temperature) adequately. This situation occurs when the statistical model adapts too much to the training data (overfitting). To test the sensitivity of the RFM to the training data, 50 runoff reconstructions are generated using a Monte Carlo approach in which the RFM is trained using a random 60%-subset of the grid-cells with observations. The ensemble mean of the realizations is referred, hereinafter, as the GRUN reconstruction (Ghiggi et al., 2019). The ensemble of realizations is in turn used to investigate the model sensitivity to the training data at multiple spatio-temporal scales in Sect. 5.4.

## 4.3 Model Validation

### 4.3.1 Cross-validation at the grid-cell scale

Within a cross-validation framework (Hastie et al., 2009), the available data are split in a training and test set. Training data are used to build the statistical model, while the test data are employed to assess the ability of the algorithm to predict information unavailable during the training process. To evaluate the agreement of runoff predictions with observations, two target-oriented (Meyer et al., 2018) cross-validation (CV) experiments named CV-SREX and CV-SPACE are set up, which avoid an over-optimistic view of model performance.

CV-SREX aims to evaluate the ability of the model to extrapolate in the situation where no nearby runoff observations are available at all. For this purpose, the globe is divided in 26 subcontinental regions (Fig. 4) as defined in the Special Report for EXtremes (SREX) of the Intergovernmental Panel for Climate Change (Seneviratne et al., 2012). Successively, at each cross-validation step, all observations within a SREX region are removed from the training dataset and subsequently used to test the performance of the RFM. This implies that the rainfall-runoff relationship is learned and transferred from regions far away as local information is not available to calibrate the model.

CV-SPACE follows the approach of GS15,16 and aims to assess the effective prediction accuracy in data-rich regions, where nearby runoff observations can provide information to refine the runoff estimates. In this case, the grid-cells are randomly divided into ten folds. Then, at each cross-validation step, one fold is used as test set, while the observations of the remaining folds are used as training data.



### 4.3.2 Validation at the basin-scale

The selection of 214 GRDC river discharge observations detailed in Sect. 3.3 is used to assess the accuracy of the GRUN reconstruction in large river basin (area larger than 50'000 km$^2$). GRUN-based first-order river discharge estimates are obtained by spatially averaging the grid-cell runoff times series within the basin and multiplying by the drainage area. At monthly time scale, the effect of water routing is considered negligible for most of basins.

### 4.3.3 Performance Metrics

Seven performance metrics are employed to assess the accuracy of the RFM in reproducing different aspects of the runoff time series. Model skill is determined for each cross-validated grid-cell and for each selected large GRDC river basin. The terms $p_t$ and $o_t$ refer to the predicted and observed time series respectively.

1. The relative bias (*relBIAS*) has an optimal value of 0 and allows to investigate the presence of systematic errors. A positive (negative) value indicates a general overestimation (underestimation). It is defined as:

$$relBIAS = \frac{mean(p_t - o_t)}{mean(o_t)} \quad (1)$$

2. The ratio of standard deviations (*rSD*) has an optimal value of 1. Values lower than 1 indicate underestimation, while values higher than 1 indicate overestimation of the observed variability. It is defined as:

$$rSD = \frac{sd(p_t)}{sd(o_t)} \quad (2)$$

3. The squared correlation coefficient, $R2$, ranges between 0 and 1. It measures the degree of the linear association between the predicted time series and the observed ones. It is insensitive to the bias. The optimal value is 1.

4. The Nash Sutcliffe Efficiency (*NSE*), also called model efficiency, (Nash and Sutcliffe, 1970) is a measure of the overall skill of the model. *NSE* = 1 corresponds to a perfect match between predicted and observed data, while a value lower than 0 indicates that model predictions are on average less accurate than using the mean of the observed data. It is defined as:

$$NSE = 1 - \frac{\sum_t (p_t - o_t)^2}{\sum_t (o_t - mean(o_t))^2} \quad (3)$$

where $mean(o_t)$ refers to the long-term mean of the observations.

5. The squared correlation coefficient between the observed and predicted monthly standardized anomalies (i.e. monthly time series with the monthly climatology removed, divided by the long-term standard deviation of each month), $R2_{ANOM}$. It ranges from 0 to 1 (best value).

6. The squared correlation coefficient between the observed and predicted monthly climatology, $R2_{Clim}$. It ranges between 0 and 1 (best values).



## 5 Evaluation of the runoff reconstruction

### 5.1 Grid-cell scale validation

To evaluate the validity of the runoff reconstruction at different time scales, Fig. 3 reports scatterplots between observations and the CV-SPACE predictions for monthly, annual and long-term mean values. Overall the agreement is satisfactory,

although there is a tendency to underestimate runoff rates when the magnitude increases.

Figure 4 shows the spatial distribution of the considered skill scores based on the two cross-validation experiments CV-SREX and CV-SPACE, while Table 1 reports the median values of the grid-cell skill scores distribution. The spatial patterns emerging from the two cross-validation experiments are very similar, with CV-SPACE displaying better scores because of the RFM ability to exploit local information to improve runoff estimates.

On average, the *relBIAS* of the RFM is slightly negative, indicating a tendency to underestimate monthly runoff rates (Fig. 4a-b). However, in arid regions such Southwest USA, Northeast Brazil and Southern Africa, the RFM tends to overestimate the runoff (*relBIAS* is positive). Figure 4c-d show that when runoff is overestimated also the variability tends to be exaggerated (*rSD* is higher than 1). Oppositely, in the others area, the variability is generally underestimated.

Overall, the runoff dynamics are well reproduced as indicated by high values of $R2$ (Fig. 4e-f). The *NSE* skill scores (Fig.

5g-h) shows that in most regions of the world, RFM predictions are more skillful than the observed runoff long-term mean ($NSE > 0$). The accuracy in reproducing runoff anomalies show a more complex spatial pattern (Fig. 4i-j): humid regions and lowlands have quite high $R2_{anom}$ values, while decreasing skill is observed in mountainous regions and arid regions.

Finally, Fig. 4k-l illustrate that the seasonal cycle of runoff is excellently reproduced across all the globe.

### 5.2 Basin-scale validation

Figure 5a illustrates the accuracy of GRUN using the selection of GRDC reference streamflow stations, while Fig. 5b shows the observational agreement of river flow timeseries for selected basins displayed in Fig. 5a. The temporal evolution of river flow is in general well captured and an underestimation of the peak flow volume is only evident for the Mekong river. For the Ebro the agreement between observations and GRUN start to decrease from 1965 ahead. The dynamics are no more well captured, and GRUN estimates are constantly higher than the GRDC observations. These discrepancies might be caused by

the intensive irrigation and reservoir activities which have altered the natural hydrological regime of the basin. In that respect, it is interesting to notice that the NSE spatial pattern in Fig. 5a show many similitudes with the estimated amount of runoff stored by engineered impoundments reported in Vörösmarty et al. (2004): low NSE scores tends to correspond to higher fractions of water impoundment. Both the Nile and Colorado river basins are an exceptional example of the human-induced river flow alterations.

However, human activities are not the only cause of discrepancies between GRUN-based river discharge estimates and the observations. In the Amazon river, the negative NSE value and the visible phase lag between the estimated and the observed time series is not caused by an inaccurate runoff reconstruction, but rather related the fact that river discharge is simply



estimated using the average runoff within the basin. Indeed, for such a very large river basin, a routing model accounting for water travel would be necessary to correctly reproduce the river flow dynamics also at monthly timescales.

## 5.3 Benchmarking against global hydrological models

To benchmark the performance of GRUN against well-established GHMs, the skill of the RFM is compared to the skill of 13
GHMs runoff simulations. Figure 6 compares the distribution of the skill scores for the CV-SREX and CV-SPACE experiments against the skill of the ISIMIP2a GHMs runoff simulations.

CV-SPACE always has higher skills than CV-SREX and outperforms all ISIMIP2a GHMs runoff simulations and their multi-model ensemble mean (MMM). CV-SREX outperform the MMM except for *relBIAS* and $R2_{anom}$.

Overall, the GRUN cross-validation experiments show a tendency to underestimate runoff although the spread of relBIAS is
reduced compared to the ISIMIP2a models. Among the GHMs there is not a clear tendency to under/overestimate runoff. The same applies for the variability (*rSD*). The dynamics of runoff (*R2*) are much better reproduced by GRUN than the considered GHMs, and the overall *NSE* skill score distribution is much better for GRUN than for the ISIMIP2a GHMs simulations.  The anomalies ($R2_{anom}$) are also much better reproduced by GRUN, and CV-SREX outperforms all the single GHMs.

Finally, $R2_{clim}$ demonstrates that GRUN reproduces much better the seasonal cycle than the GHMs. Previous studies already showed that GHMs struggle in reproducing the seasonality of runoff (Gudmundsson et al., 2012; Gudmundsson and Seneviratne, 2015).

## 5.4 Sensitivity of the runoff estimates to the observations used for training

An ensemble of 50 runoff reconstructions trained on different subset of observations (Sect. 4.2) is used to assess the
sensitivity of GRUN to the observations used for training. Figure 7 shows the long-term mean of the monthly ensemble standard deviation and coefficient of variation (defined as the standard deviation divided by the mean). Regions characterized by higher runoff rates shows higher standard deviation (Fig. 7a), but this variability across the realizations is small (< 20 %) compared to the runoff magnitude (Fig. 7b). Except of arid regions, the coefficient of variation is generally below 0.2 (Fig. 7b).

To put into perspective the sensitivity of GRUN to the observations used for training, Figure 8 compares the annual runoff volumes of the GRUN realizations against the state-of-the-art GHMs participating in ISIMIP2a.

The global long-term mean runoff volume estimated by GRUN (37419 km³/yr) lies within the lower range of ISIMIP2a GHMs (Fig. 8a) and closely agrees with others global terrestrial discharge estimates (Table 3). The uncertainty attributed to the selection of training observations (shaded area in Fig 8a) of the global GRUN runoff volume is far smaller than the
spread introduced by different physical representations of the hydrological processes in the GHMs. The uncertainty introduced by the selection of training observations increases proportionally with the magnitude of the runoff rates and is highest in the tropics (Fig. 8b). Reversely, the spread of the GHMs tends to be constant across all latitudes. GRUN has



almost always latitudinal mean runoff rates lower than the MMM but goes beyond the GHMs range only between 20° and 30° latitudes North. This pattern is related to the relatively low runoff estimates in GRUN in Northeast India compared to the GHMs (Fig. 8c).

**5.5 Limitations of GRUN**

The streamflow observations used for model training underwent careful preprocessing and screening steps to remove timeseries presenting sudden changes in the hydrological signature. Therefore, and because solely forced with precipitation and temperature, GRUN do not account for the effects of human river flow regulation on the reconstructed hydrological regimes. However, we note that some streamflow observations impacted by irrigation and other land- and water management practices have likely not been removed, especially if the magnitude of water abstraction/returns did not alter the monthly

hydrograph sufficiently to be detected by change point detection tests. This may be one of the reasons for the overestimation of runoff rates in several arid regions (Fig 4a-b) known for intensive-irrigation activities (Wriedt et al., 2009; Siebert et al., 2015).

To some extent, the impact of past land-use changes on water availability might be implicitly accounted for in GRUN if the GSWP3 reanalysis captured the changes in precipitation and temperature which were induce by such human activities

through land-atmosphere interactions. Any changing pattern in water availability emerging from GRUN is however solely conditioned by trends of the GSWP3 forcing and the runoff observations used for model training.

Finally, the accuracy of the runoff rates in mountainous regions is likely not optimal. The spatial resolution of the considered meteorological forcing does not allow to capture the sub-grid variability of precipitation and temperature that governs e.g. snow-melt volume and timing in such regions. Glacier melting is also not accounted for in GRUN.

**6. Example Applications**

**6.1 Runoff climatology**

Figure 9a displays the annual runoff climatology derived as the long-term mean of the GRUN reconstruction covering the 1902-2014 period. Figure 9a shows that long-term mean runoff rates can differ by three orders of magnitude across the globe. The highest runoff rates are observed mostly in the tropics and in large mountain ranges. The extratropics show low

runoff rates, in correspondence with the major world deserts such as the Sahara. Monthly climatologies are provided in the supporting information (Fig. S1).

Figure 9b and 9c show the months with the maximum and minimum of the long term mean seasonal cycle. In the Northern Hemisphere, regions exposed to winter snow accumulation have the lowest runoff during the coldest months and a runoff peak toward the end of spring which is related to the melting of the snowpack. In the humid mid-latitudes,

evapotranspiration closely follows the seasonal cycle of surface temperature, causing the lowest (highest) runoff to occur prevalently during the summer (winter) months. In the tropics, the month with maximum runoff tends to occur during the



rainy season. In the Northern tropics this occurs between August and September, while in the Southern tropics between February and April, following the migration of the Intertropical Convergence Zone (Schneider et al., 2014).

## 6.2 Trends in reconstructed runoff

GRUN can be used to investigate changing freshwater availability. Figure 10 displays the annual runoff trends for the period
1971-2010 computed using Sen's slope (Sen, 1968) and expressed in absolute and relative terms. Overall the reconstructed trends are in line with other reported findings (Gudmundsson et al., 2018a) and closely resemble the observed trends (Fig. 10a).

In Europe, the Mediterranean regions exhibits a decrease in annual runoff, while in Central and Northern Europe there is a tendency to increasing runoff rates. This pattern is in agreement with previous studies (Stahl et al., 2010, 2012) and was
recently attributed to anthropogenic climate change (Gudmundsson et al., 2017). In the Eastern and Western USA negative trends occur, while large portions of the Mississippi river basin show increasing runoff.

In the tropics, the Amazon basin shows a substantial decrease in annual runoff rates and a reduction of freshwater discharge to the Atlantic Ocean has the potential to impact the Atlantic and the Northern Hemisphere climate (Vizy and Cook, 2010; Jahfer et al., 2017). In light of the human pressure to which this basin is currently exposed (Castello and Macedo, 2016;
Latrubesse et al., 2017) and the uncertain impact of deforestation on river flow (D'Almeida et al., 2007; Spracklen et al., 2012; Lawrence and Vandecar, 2015; Spracklen and Garcia-Carreras, 2015), the causes and consequences of such trends should be investigated in more detail. Similarly, the drying tendency observed in many regions of the Congo Basin could affect the Eastern Equatorial Atlantic climate variability (Materia et al., 2012). Reversely, tropical area situated in Southeast Asia experiences an increase in runoff.

The monthly resolution of GRUN also allows to investigate these changes at subseasonal time scales (Fig. S2), which might e.g. be relevant for water resource assessments because neglecting the seasonal fluctuations can cause underestimation of water scarcity (Mekonnen and Hoekstra, 2016). In addition to changes in magnitude, the GSIM dataset offers also the possibility to analyze shifts in the seasonality of the hydrological regimes. Figure S3 provide an overview of the months in which the minimum and maximum runoff volumes occurred at the beginning and at the end of the 20[th] century. Over
Europe, the pattern of change is consistent to ones emerging in recent studies (Blöschl et al., 2017; Hall and Blöschl, 2018).

## 6.3 Interannual variability and teleconnections

The long temporal coverage of GRUN combined with its high skill in reproducing runoff dynamics provides an unprecedented opportunity to study the response of runoff to the modes of climate variability throughout the 20[th] and the early 21[st] century. The Hovmöller diagram in Figure 11a illustrates the interannual runoff variability by showing the time
evolution of the latitudinal mean of monthly runoff standard anomalies. The occurrence of El Niño events, defined here as the periods in which the Multivariate ENSO Index (MEI) (Wolter and Timlin, 2011) is larger than 1, coincides with negative anomalies in the tropical regions. A correlation analysis between monthly standard anomalies of GRUN and the MEI



timeseries reveals that during El Niño events, the Amazon basin, the Southeast Asia, Australia and South Africa tend to experience lower runoff rates (Fig. 11c), which is consistent with previous assessments (Ward et al., 2010; Wanders and Wada, 2015). The opposite occurs during la Niña conditions, and drier conditions are observed in the western United States, which is also consistent with previous work (Tang et al., 2016).

As an additional example, Fig. 11d shows the influence of the North Atlantic Oscillation (NAO) on the European continent. The analysis confirms the previous finding that when NAO is positive, England and Scandinavian exhibit higher runoff rates, while Southern Europe experiences drier conditions (Bouwer et al., 2006; Bierkens and van Beek, 2009; Lorenzo-Lacruz et al., 2011; Steirou et al., 2017).

## 6.4 Drought and agricultural productivity

GRUN can be exploited to study the spatio-temporal development of slowly evolving phenomena such as droughts. Since runoff can be defined as the excess of water available to ecosystems, negative runoff standard anomalies can be used as an indicator for droughts and potentially lower agricultural productivity (GS15, GS16). Figure 12 shows three drought events that are known for their exceptionally severity and devastating impact on agricultural production. Figure 12a displays runoff standard anomalies over Europe for the month of August in 1976, which, according to our results, ranks in the top 5 driest

months for the entire period 1902-2014 in large parts of England, Northern France, Central Europe and Southern Sweden. Studies have shown that the drought mainly developed because of severe precipitation deficits (Zaidman et al., 2002; Spinoni et al., 2015) rather than extremely hot temperature such as during the 2015 drought (Ionita et al., 2017).

Figure 12b reports the annual runoff standard anomalies in North America for the year 1934. This drought is known as the "Dust Bowl" and is unique for its spatial extent and duration. The negative runoff anomalies span the entire United States.

Several studies have suggested that initial drying caused by La Nina conditions was amplified by human-induced land degradation of the US Plains (Schubert et al., 2004; Cook et al., 2009, 2014). During this event, dust storms severely damaged the American prairies by destroying million hectares of cultivated land. Finally, Fig. 12c illustrates the Horn of Africa drought conditions in 1984. The event also ranks in the regional most extreme events and resulted in a widespread famine which killed as much as 700'000 people in Ethiopia (Kidane, 1990). The drought was linked to El Nino conditions

and a strong reduction in annual precipitation (Viste et al., 2013; Lanckriet et al., 2015).

## 7. Conclusion and outlook

This study presents an observationally-driven global gridded reconstruction of monthly runoff rates derived using a machine learning algorithm. The dataset covers the period from 1902 to 2014 and is provided on a 0.5 x 0.5° WGS84 grid.

A machine learning algorithm is trained with runoff observations from a global collection of in-situ streamflow observations

of relatively small catchments (< 2500 km$^2$) and uses gridded precipitation and temperature from a century-long reanalysis product as predictors. Model validation based on cross-validation experiments shows that the accuracy of the reconstruction

is reasonable. The reconstruction has on average a higher skill than a collection of state-of-the-art global hydrological models, especially with respect to the reproduction of the seasonality, dynamics and anomalies of runoff. At the monthly time scale, we find that a restricted number of predictors (i.e. precipitation and temperature) is sufficient to reproduce important aspects of terrestrial water dynamics. GRUN is thus an interesting candidate to evaluate and refine current

parametrizations of the global hydrological models as well as to potentially constrain fluxes of fine-resolution models (in space and time) throughout the adoption of multiscale optimization techniques (Samaniego et al., 2010, 2017).

Since GRUN does not account for the impact of human river flow regulation and land-use change, differences between this reconstruction and in-situ observations could potentially be used to quantify the impact of flow regulation at a regional to global scale (Jaramillo and Destouni, 2015; Arheimer et al., 2017), as well as to systematically identify streamflow stations

which have a hydrological regime very different from the naturalized flow as predicted by GRUN.

Finally, GRUN offers a unique view on large-scale features of runoff variability also in regions with limited or no observational coverage. The new dataset can be exploited (i) to study the onset and development of large-scale extreme events such droughts, (ii) to investigate links between runoff and modes of climate variability, (iii) to conduct large-scale water resources assessments, (iv) to detect changes in water availability and dynamics and (v) to address other new scientific

challenges in water cycle research (Wagener et al., 2010; Montanari et al., 2013; Greve et al., 2014; Trenberth and Asrar, 2014; Hegerl et al., 2015).

We conclude by remarking that this dataset would not have been possible without the mobilization of national and international hydrological archives. This study shows the benefit of a wider access to hydrological data collected by various institutions worldwide. We call for a continuation of the international efforts to reduce political and technical barriers for the

exchange of hydrometeorological data across the scientific community.

## 8. Data availability

The GRUN reconstruction based on GSWP3 forcing is publicly available in NetCDF-4 format (Ghiggi et al., 2019) and can be freely downloaded from the ETHZ Research Collection (https://doi.org/10.3929/ethz-b-000324386).

## 9. Competing interests

The authors declare that they have no conflict of interest.

## 10. Author contribution

LG initiated this investigation. GG, VH, SIS and LG designed the study. GG developed the model code and performed the analysis. GG prepared the manuscript with contributions from all co-authors.





## 11. Acknowledgments

We thank Prof. Dr. Hyungjun Kim for providing us with early access to the GSWP3 dataset and GRDC for the river discharge observations. SIS acknowledges partial support from the ERC DROUGHT-HEAT project funded by the European Community's Seventh Framework Programme (grant agreement FP7-IDEAS-ERC-617518).

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





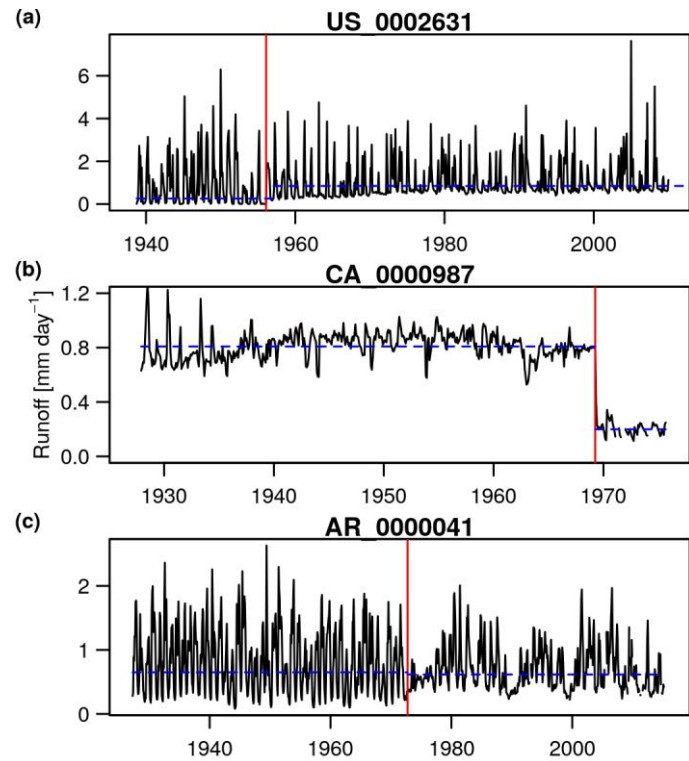

**Figure 1.** Detection of change points in runoff time series. The vertical red line indicates the change point in variance detected by the univariate normal change point in variance test, while the horizontal blue dashed lines illustrate the change in mean identified by the test in univariate normal change point in normal mean and variance. The title of the individual panels corresponds to the station identifier as used in GSIM.



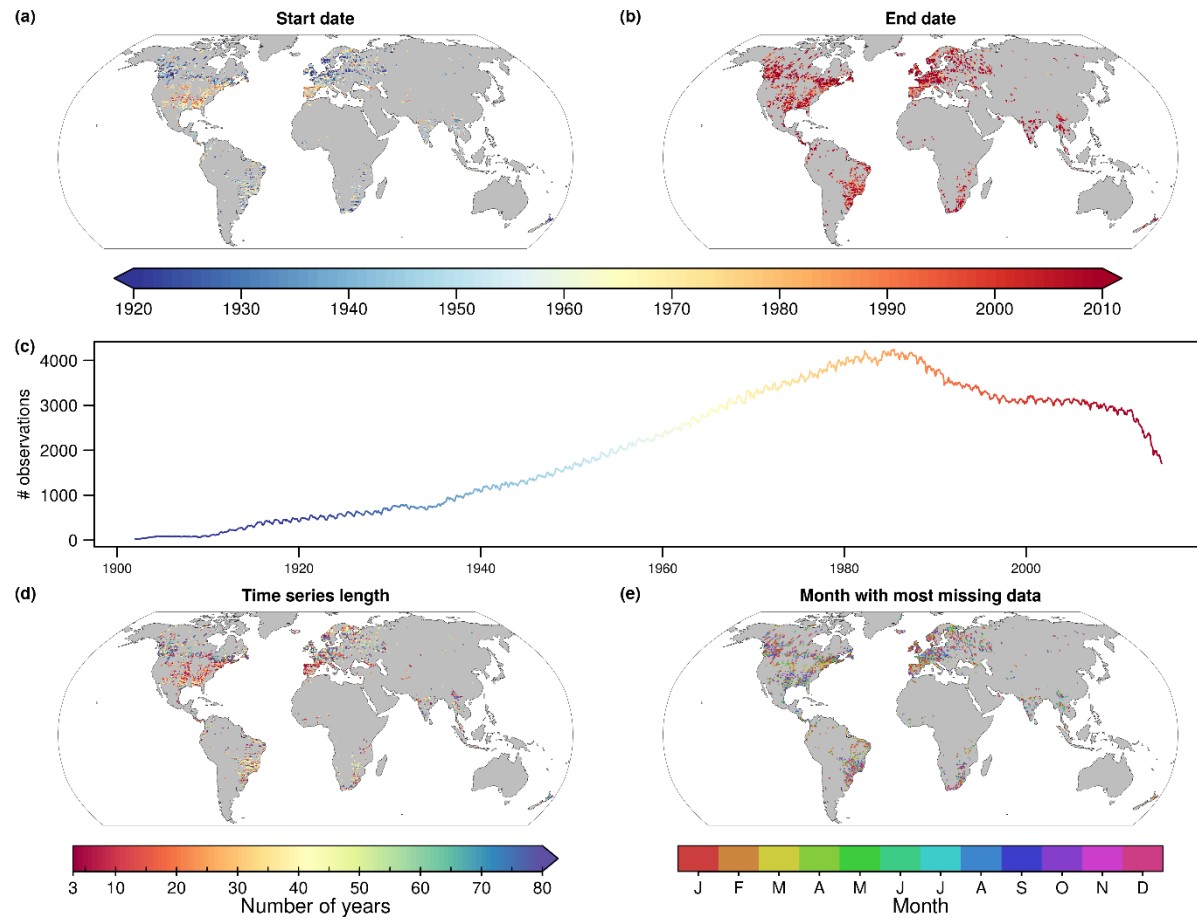

**Figure 2.** Spatio-temporal coverage of grid-cell runoff observations. a) and b) display the start and end year of the timeseries respectively. c) Total number of runoff observations at each month between 1902 and 2014. d) Numbers of years with at least 10 runoff observations in each year. e) Month with most missing values.

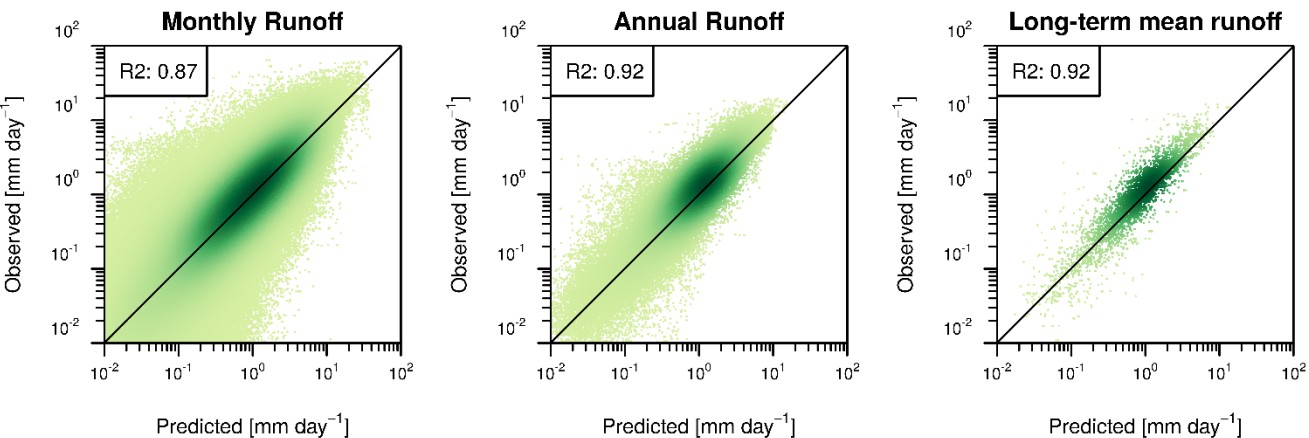

**Figure 3:** Scatterplot of observed versus predicted runoff values. The color intensity is related to the points density.





**Figure 4:** Spatial distribution of the skills scores obtained from CV-SREX (left) and CV-SPACE (right) experiments. SREX region boundaries are superimposed over the skill maps of CV-SREX.



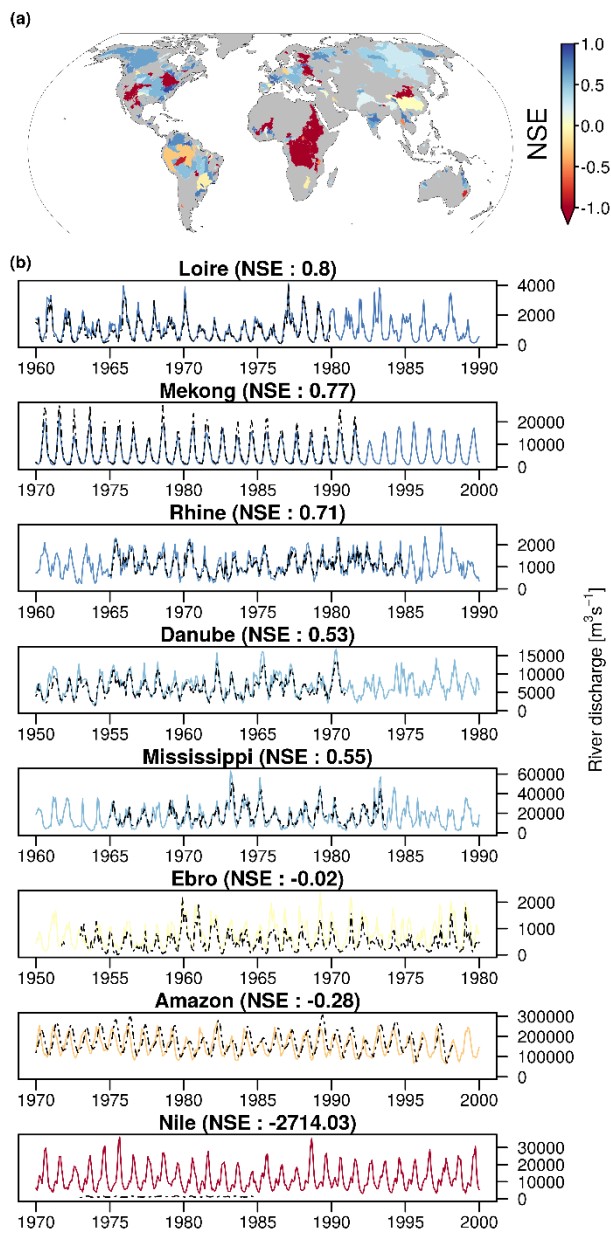

**Figure 5:** Validation results based on selected GRDC river discharge observations. a) Spatial distribution of the NSE skill for selected GRDC large basins. b) Observed (dashed black line) and predicted (colored) river discharge time series. Line colours correspond to the NSE skill shown in panel a.





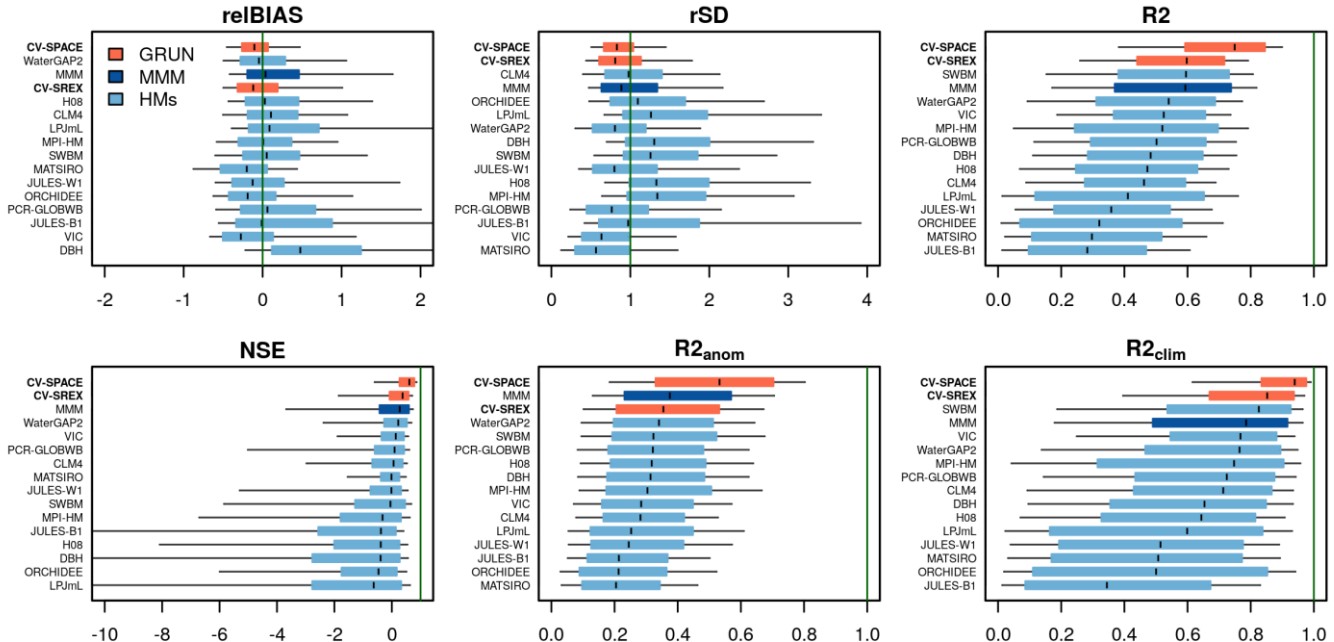

**Figure 6:** Benchmarking the performance of GRUN against ISIMIP2a GHMs runoff simulations. Boxplot whiskers cover the 0.1 to 0.9 quantiles of the skill score distribution. The dark green vertical lines indicate the optimal score. GRUN cross-validation results are displayed in orange, while the multi model mean (MMM) of ISIMIP2a GHMs runoff simulations is displayed in dark blue. In most of the cases, the order of the boxes follows the rank of the median skill score. However, to avoid compensatory effect with relBIAS and rSD scores, the individual boxes are ranked based on the median absolute value of the skill score minus the optimal score. The x-axis of relBIAS is left and right truncated, of rSD it is right truncated and for NSE it is left truncated.

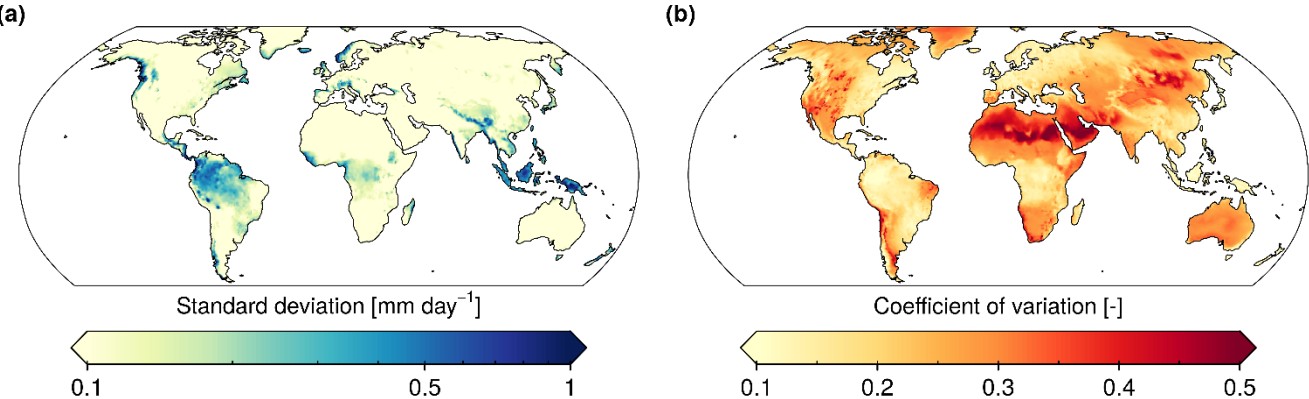

**Figure 7.** Long-term mean of the monthly standard deviation of the runoff reconstruction ensemble (a) and the corresponding coefficient of variation (b).

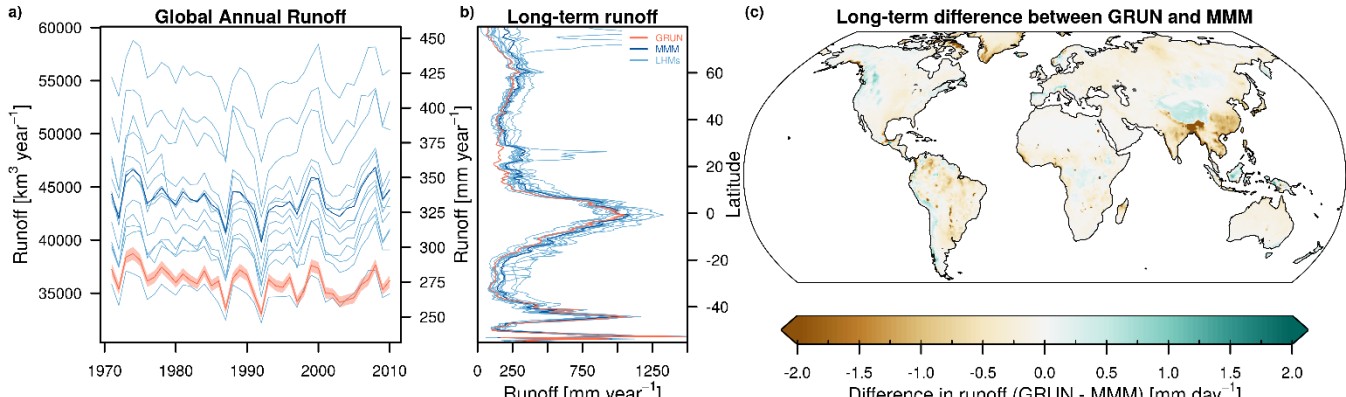

**Figure 8.** The uncertainty of GRUN, attributed to the finite sample of training data, compared to the spread introduced by different physical representations of the hydrological processes in the ISIMIP2a GHMs. The shaded area around GRUN lines shows the 10 and 90 percentiles of the GRUN ensemble distribution. a) Global annual runoff b) Latitudinal average of long-term mean runoff c) Difference between GRUN and the MMM long-term mean runoff. Grey cells represent missing values caused by missing data in some of the ISIMIP2a GHMs simulations.



**Figure 9:** Runoff climatology (1902-2014). a) Long-term mean annual runoff rates. b) Month with the minimum and c) the maximum long-term mean monthly runoff.





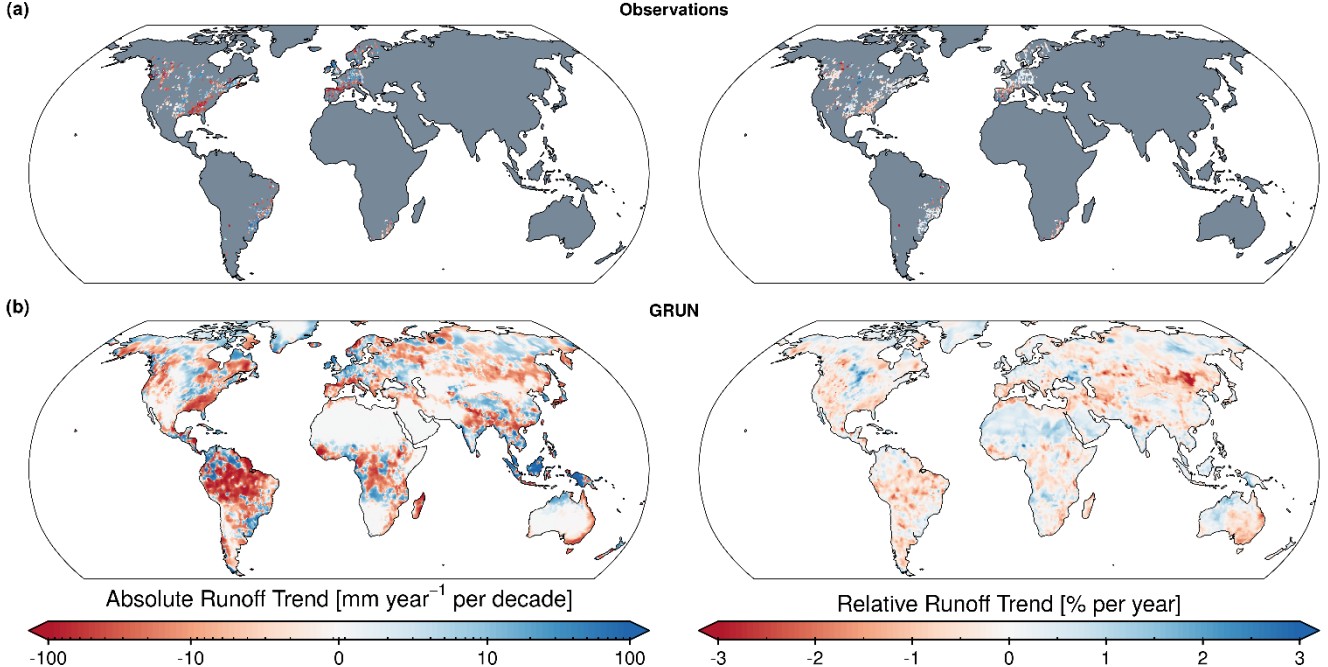

**Figure 10:** Changes in annual runoff rates (1971-2010) expressed in absolute terms and percentage change relative to long-term mean. a) Trends based on observations. b) Trends based on GRUN.



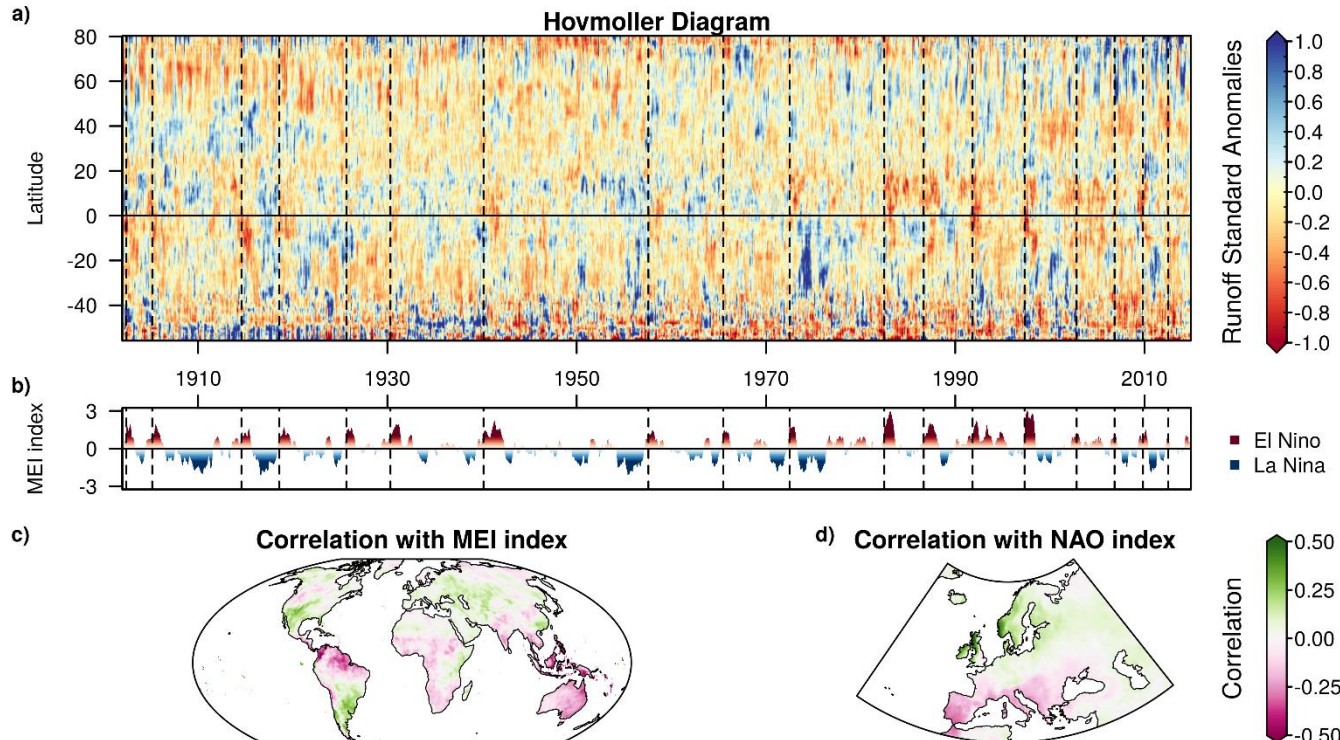

**Figure 11:** Interannual variability of runoff and its relation to modes of climate variability. a) Hovmöller diagram of standard runoff anomalies (reference period 1902-2014). Vertical dashed lines indicate onset of El Nino events. b) Timeseries of the Multivariate ENSO Index (MEI). Red and blue shades characterize the intensity of El Nino and La Nina conditions respectively. c) Correlation of the MEI with monthly runoff anomalies. d) Relationship of European runoff anomalies with the North Atlantic Oscillation (NAO).




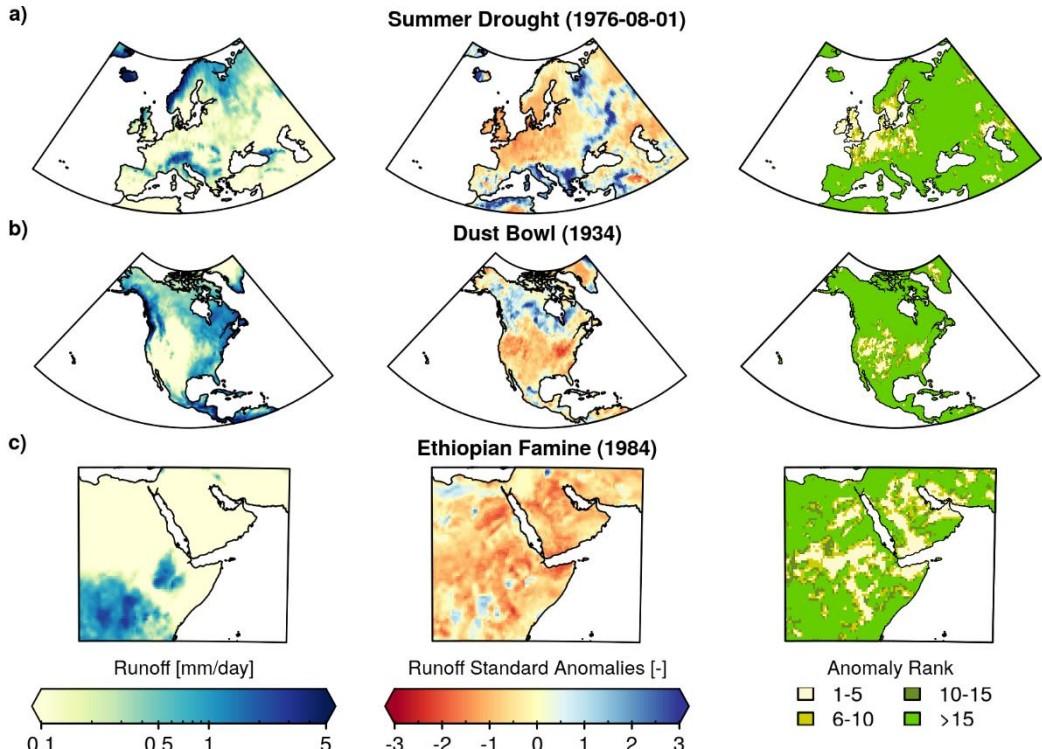

**Figure 12:** Three extreme drought events. a) Dust Bowl b) European summer drought in 1976 c) Ethiopian famine in 1984.



**Table 1.** Median values of skill scores for CV-SREX and CV-SPACE

|  | CV-SREX | CV-SPACE |
|---|---|---|
| relBIAS | -0.138 | -0.116 |
| rSD | 0.781 | 0.807 |
| R2 | 0.577 | 0.725 |
| NSE | 0.366 | 0.585 |
| R2$_{anom}$ | 0.358 | 0.556 |
| R2$_{clim}$ | 0.858 | 0.941 |

5 **Table 2.** Median values of skill scores computed for the large GRDC river basins

| relBIAS | rSD | R2 | NSE | R2$_{anom}$ | R2$_{clim}$ |
|---|---|---|---|---|---|
| 0.009 | 1.017 | 0.732 | 0.503 | 0.437 | 0.908 |

**Table 3.** Comparing global long-term mean runoff from GRUN against values reported in the literature. GRUN estimates are obtained by considering the same time span and spatial coverage of the reported studies. Values in parenthesis denote the

10 uncertainty range reported in some studies.

| Reference | Runoff (km$^3$ yr$^{-1}$) | | Time period | Greenland | Antarctica |
|---|---|---|---|---|---|
| | Reference | GRUN[b] | | | |
| Dai and Trenberth, 2002[a] | 37288 | - | - | No | No |
| Fekete et al., 2002 | 39319 | - | "Climatology" | No | No |
| Döll et al., 2003 | 36687 | 38095 | 1961-1990 | Yes | No |
| Syed et al., 2009[a] | 30354 | 35536 | 2003-2005 | No | No |
| Wisser et al., 2010 | 37984 | 37724[c] | 1901-2002 | Yes | No |
| WaterMIP (Haddeland et al., 2011) | 42000-66000[d] | 36962 | 1985-1999 | No | No |
| Clark et al., 2015[a] | 44200 | 37865 | 1950-2008 | Yes | Yes |
| Rodell et al., 2015[a] | 45900 | 37316 | 2000-2010 | Yes | Yes |
| Müller Schmied et al., 2016 | 41298 (39200-42200) | 37590 | 1971-2001 | No | No |
| eartH2Observe (Schellekens et al., 2017) | 46268 (38652-55877) | 37137 | 1979-2012 | No | No |
| ISIMIP2a (GSWP3, nosoc) | 45180 (35997-57323) | 37419 | 1971-2010 | No | No |

[a]The long-term mean is obtained by extrapolation from continental-scale river-discharge observations or water balance.
[b]Antarctica is never included in the GRUN estimate of the global long-term mean runoff. Greenland is considered only if included in the reference dataset.
[c]GRUN long-term mean runoff is computed for the period 1902-2002.
15 [d]Haddeland et al. (2011) report that the CRU land mask used to rescale global mean runoff (excluding Antarctica and Greenland) has an area of 1.44·10$^8$ km$^2$, while the correct area value should range around 1.33·10$^8$ km$^2$.