# Peer review of "GRUN: An observations-based global gridded runoff dataset from 1902 to 2014"

_Earth System Science Data, 2019_

## Referee Comment (RC1) · Anonymous Referee #1 · 16 Apr 2019

This study, "GRUN: An observations-based global gridded runoff dataset from 1902 to 2014," presents a new global dataset of monthly runoff fields from 1902 to 2014, as the title says. The runoff fields are produced by upscaling the observed discharge in small river basins (smaller than a grid cell) using the time series of monthly precipitation and temperature, and their time lag operators.

I find the data product timely (actually, well overdue) and very relevant for the evaluation of global scale land surface and hydrological models. The paper is well-written with clear explanations of the methods and data. Additionally, it includes some suitable examples/case studies, and shows that the data is able to capture the main features discussed therein.

I, however, have several concerns and suggestions (see below) which should be addressed before the manuscript is published. The additional analysis and discussion will strengthen the data product.

**Data and Methods:**

- The area of basins selected for fitting the random forest model is well below the size of a grid cell. That means that the discharge in these basins might be driven by sub-grid precipitation (and climate) that is not reflected at the grid-scale. As a sanity check, one should compare the observed discharge against the gridded precipitation. Perhaps, even filter out the basins where the annual discharge is higher than annual precipitation? If not, the results should show the distribution of discharge over precipitations in basins with observed discharge.

- The study would benefit by including a figure showing/comparing the distribution of data values in whole and training dataset. It is known that random forest cannot predict outside the range of data values used in training.

- There is no explanation on why the predictors included precipitation and temperature only, especially considering that the co-authors have produced a long-time series of terrestrial water storage variations at the global scale. In addition, several other factors such as vegetation properties, topographic indices, soil properties, and so on contain useful information on runoff generation mechanisms. Previously, these variables have been shown to include information on baseflow characteristics globally (see H. Beck et al, 2013 (WRR), and 2015 (JHM)).

**Data Evaluation:**

- The validation of the predicted runoff against GRDC should include the comparison of mean flow in addition to the time series (and metrics based on it). The mean discharge and runoff over the basin should be equal if there is no long-term change in river storage. If the mean runoff from GRUN is similar to GRDC observed discharge, one can infer that the poorer performance in the time series is due to exclusion of river routing.

- Is it possible that the poorer performance in the humid basin is related to lack of data values similar to those in Amazon and Congo in the training dataset? Additionally, in the comparison of global volumes, the GRUN also falls in the lower range. I am curious if this underprediction is coming from the lower values in the tropics which dominate the contribution to global runoff. This can be checked by having a grid-to-grid comparison (something like hexbins in Figure 3) against previous products or model simulations. If the underprediction is clear, it makes me wonder if predicting the runoff ratio (runoff to precipitation) would be beneficial. A discussion on this would help the future studies on a similar topic.

- The GRDC discharge observations are likely to be affected by anthropogenic uses, as the manuscript rightly points out several times. Therefore, wouldn't it make sense to evaluate the GRUN product against:

    o previous mean total runoff from Beck et al., 2015 which uses fewer observation stations but more predictors (also based on neural network),

    o mean ET from, say, LandFlux-EVAL, part of which is coming from satellite observations?

**References:**

Beck, Hylke E., et al. "Global patterns in base flow index and recession based on streamflow observations from 3394 catchments." *Water Resources Research* 49.12 (2013): 7843-7863.

Beck, Hylke E., Ad De Roo, and Albert IJM van Dijk. "Global maps of streamflow characteristics based on observations from several thousand catchments." *Journal of Hydrometeorology* 16.4 (2015): 1478-1501.

---

## Referee Comment (RC2) · Anonymous Referee #2 · 21 Jun 2019

The manuscript by Ghiggi et al. presents an extension of the methodology developed by Gudmundsson and Seneviratne (2015,2016) in order to reconstruct monthly runoff rates at global scale at 0.5 deg spatial resolution. Runoff estimates from a large number of small catchments are combined with state-of-the-art atmospheric reanalysis to train a machine learning algorithm (random forests) to predict monthly runoff rates from monthly temperature and precipitation. Runoff estimates are extensively evaluated through cross-validation as well as against independent observed discharge time series and runoff estimates from global hydrological models. The study is technically sound and the manuscript is clearly written. The presented data-set is the first of its kind at global scale.

Main comments:

[Figure]

The authors chose to develop the runoff model from precipitation and temperature, while GS15 found for Europe that this simple model is outperformed by one considering full atmospheric forcing - which is available through GSWP3. This choice must be justified in the methods section. Further, GS15 showed that including land surface parameters (soil texture and slope) does not improve model performance. I can only assume that this is the reason why land surface parameters were not included in this study. This choice needs to be clearly mentioned in the manuscript and it should at least be discussed whether this assumption is also valid on global scale.

GRUN runoff estimates are compared to those from an ensemble of global hydrological models. The ensemble actually consists of global hydrology and water resources models, land surface models and dynamic vegetation models. Neither type was specifically developed or is trained/calibrated to accurately predict runoff at grid cell level - and this is without question where the strength of GRUN lies. This needs to be discussed in a revised manuscript. GHMs have mainly been validated against discharge observations from catchments >10000 km2, acknowledging the uncertainty in runoff estimates at grid cell level originating from model structure and parameters, climate input, and land surface properties. To make for a fair comparison, I challenge the authors to repeat the validation experiment in section 5.3 for the GRDC discharge records. Since the authors use results from the ISIMIP nosoc experiment, catchments heavily impacted by flow regulation and/or water abstraction need to be excluded from this analysis.

The authors argue that GRUN estimates do not include human interference on runoff generation, thus that differences between the GRUN reconstruction and in-situ observations could potentially be used to identify streamflow stations which have a hydrological regime very different from the naturalized flow (P14, L9-10). I strongly disagree given that the screening procedures used are capable of identifying break points in the discharges records caused by the installment of (large) water infrastructure, e.g. dams and diversions, while they likely miss more gradual changes. Consequently, the discharge time series used for training the model are a blend of near natural catchments

and those impacted by human activity, and it is unclear to which degree human impact is actually implicitly represented in the reconstruction. This needs to be clearly stated and discussed in the revised manuscript.

The extrapolative power of GRUN is somewhat overstated (in particular P14 L7-16). The developed RF model is (in the best case) capable of predicting short-term changes in monthly runoff/water availability as a function of (changes in) precipitation and temperature with all other boundary conditions constant. Changes in water availability originate from a variety of drivers, e.g. water abstraction/diversion, land use/cover change, reservoir management, and in many cases climate may not be the most important one (see e.g. Haddeland et al., 2014). While I do not question the usability and added value of GRUN in the first place, the authors need to point out these limitations very clearly in a revised manuscript.

The authors evaluate the performance of GRUN in a cross-validation exercise. The corresponding maps displaying six performance metrics show clear regional differences in model performance, while numbers are only provided on global level (Table 1). In order to provide a more detailed picture of model performance for potential users of the data-set, I recommend to report numbers (Q5,Q50,Q95) at sub-global level, at least for the CV-SPACE experiment. This could be climate regions or the SREX regions already used by the authors.

Minor and technical comments:

P4 L17 "Climatological runoff signature" Clarify term or rephrase.

P4, L19-20 Clarify how false zero values were distinguished from e.g. dry out during drought periods

P4, L29 Hydrological signatures are index values to describe hydrological behavior, i.e. there is no single hydrological signature of a discharge time series. Please rephrase.

P5, L4 Correct to "displays"

P5, L5 Rephrase "sudden drops", e.g. step changes in the mean

P5, L1 Correct to "with an area"

P6, L4 "streamflow records heavily impacted by humans"

P6, L25 Correct to "that influence"

P6, L5 Consider rewording to "is assumed to be negligible" since routing is indeed not negligible under certain circumstances as later shown by the authors.

P8, L7 Correct to "Six performance metrics"

P10, L22 Correct to "show higher"

Sect. 6.1 The description of global runoff patterns is somewhat lengthy since it presents rather common knowledge, please shorten.

P12, L25 Which patterns are the authors referring to? Please explain in more detail.

P13, L14-15 Top 5 driest months in terms of runoff anomaly? Please clarify.

P14, L14-15 "to address other new scientific challenges in water cycle research" Either give specific examples or remove.

Fig 2: The resolution of all maps needs to be increased. In addition, I'd recommend to remove the continent outline since it blurs with the observation points.

Fig 3: Consider swapping x and y axis to ease readability and add name of the experiment to the caption. Add results from the CV-SREX experiment to the figure.

Fig 4: The map resolution needs to be improved. Please remove continent outlines to improve readability.

Fig 5b: I recommend to plot all simulated hydrographs in the same color, since the color coding according the NSE value compromises the readability of the plots.

Fig 8: Please increase resolution. The legend lists "LSMs" while the term "GHMs" is

used throughout the manuscript. In Fig 8a, consider showing the full range of GRUN ensemble simulations - since the same is done for the GHM ensemble.

Fig. 10a: Please change background color since it is hardly distinguishable from the blues in the color scale.

Tab. 1+2: Please add the 5th and 95th quantile for each performance metric.

References:

Haddeland, Ingjerd, et al. "Global water resources affected by human interventions and climate change." Proceedings of the National Academy of Sciences 111.9 (2014): 3251-3256.

---

## Author Comment (AC1) · 18 Jul 2019

We would like to thank the anonymous reviewer for the open and overall positive evaluation of our manuscript. In the following, we provide point-by-point answers to the reviewer's comments. For the sake of clarity, we first repeat the reviewer's comments (in italic) and then provide our response.

**Comment 1:**

*The area of basins selected for fitting the random forest model is well below the size of a grid cell. That means that the discharge in these basins might be driven by sub-grid precipitation (and climate) that is not reflected at the grid-scale.*

**Reply 1:**

We agree with the reviewer that the discharge at any catchment represents the average runoff response of smaller subdomains, all affected by local variability in weather, climate and topography. However, the availability of global weather observations also constitutes a limiting factor in a global study, i.e. the effective resolution of the considered atmospheric data may be even coarser in regions with limited data availability. In addition we would like to point out that, we model monthly averages that have much less sub-grid variability compared to hourly or daily time scales. We also note that the model is not trained on individual catchment responses but on the mean response over a given grid cell (averaged over the catchments falling within that grid cell, see section 3.2). It is therefore our evaluation that although sub-grid variability may contribute to estimation uncertainty, this is less critical than other uncertainties (e.g. related to the quality to the atmospheric data) in the context of our investigation.

**Comment 2:**

*As a sanity check, one should compare the observed discharge against the gridded precipitation. Perhaps, even filter out the basins where the annual discharge is higher than annual precipitation? If not, the results should show the distribution of discharge over precipitations in basins with observed discharge.*

**Reply 2:**

Many thanks for suggesting to check the physical consistency of the runoff data and the precipitation product used to create the estimates. In the vast majority of cases, the modeled annual discharge is lower than the annual precipitation. However, while such comparisons are principally informative and help to gauge the limitations of the resulting data-product, care needs to be taken when interpreting the results. For example, both the considered streamflow observations as well as the employed precipitation data may contain biases, which might be implicitly corrected by the statistical model. As a result, there may be instances where long term mean runoff exceeds precipitation: see e.g. Fig. 1 in Gudmundsson et al. (2012) for such a comparison and the associated discussion related to underestimation of precipitation in regions with complex topography. The figure below shows the grid cell locations with a runoff ratio (Q/P) larger than one for a) the training dataset and b) the final GRUN product. In many mountainous regions, GRUN seems to account for possible bias in either the precipitation forcing or streamflow observations, which is qualitatively in line with previous investigations on precipitation biases (Adam et al., 2006).

[Figure]

In the revised Section 5.5 ("Limitations of GRUN") we now point the readers to possible inconsistent water balance estimates in mountainous regions caused by bias in the precipitation or streamflow data used. Finally, we benchmarked GRUN against ISIMIP2a river flow simulations in large river basins. The results will be included in the revised paper and reveal that GRUN performs better in term of bias than any of the considered GHMs.

**Comment 3:**
*The study would benefit by including a figure showing/comparing the distribution of data values in whole and training dataset. It is known that random forest cannot predict outside the range of data values used in training.*
**Reply 3:**
Thank you for the suggestion. We will provide cumulative distribution functions of runoff and standard anomalies, for different Koppen-Geiger climate zones, for both CV-SREX and CV-SPACE cross-validation experiments (Fig. S6 and S7).

**Comment 4:**
*There is no explanation on why the predictors included precipitation and temperature only, especially considering that the co-authors have produced a long-time series of terrestrial water storage variations at the global scale.*
**Reply 4:**
Thank you for the interesting suggestion to also include the century long terrestrial water storage (TWS) reconstruction (GRACE-REC) (Humphrey and Gudmundsson, 2019) as forcing. Note, however, that the TWS reconstruction is based on the same meteorological data as GRUN. Consequently, although the TWS reconstruction is based on a different method, our evaluation is that it does not contain any additional information that the Random Forest algorithm would be able to capture. We acknowledge that TWS is of course essential for runoff generation, and this is implicitly accounted in GRUN by including antecedent monthly precipitation and temperature in the statistical model.
The choice of the predictors will be better explained in the revised Section 4.1 ("Model setup").

**Comment 5:**
*Several other factors such as vegetation properties, topographic indices, soil properties, and so on contain useful information on runoff generation mechanisms. Previously, these variables have been shown to include information on baseflow characteristics globally (see H. Beck et al, 2013 (WRR), and 2015 (JHM)).*
**Reply 5:**
We recognize that such factors are important when estimating streamflow characteristics from sub-daily up to weekly time scales. However, in this paper, we focus on the monthly amount of water draining from grid cells of 50 km spatial resolution. In this context Gudmundsson and Seneviratne (2015) demonstrated that the performance is already high when considering P and T as drivers only and that inclusion of selected land properties did not improve the accuracy of the estimate. One reason which could explain why land properties do not improve our model predictions is that the random forest algorithm can synthesize information about the average climatic conditions governing e.g. vegetation properties from the input precipitation and temperature data, such that the margin for model improvement is already small (also see response to Comment 1 of Reviewer 2).

**Comment 6:**
*The validation of the predicted runoff against GRDC should include the comparison of mean flow in addition to the time series (and metrics based on it).*
**Reply 6:**

In the revised manuscript, we will provide spatial maps of relBIAS (under/over estimation of the mean flow), rSD, R2, NSE, $R2_{anom}$, $R2_{clim}$ for large river basins from GRDC (Fig. S8). Figures S9 to S12 will also compare river discharge estimates based on GRUN against equivalent estimates from ISIMIP2a GHMs simulations.

**Comment 7:**
*The mean discharge and runoff over the basin should be equal if there is no long-term change in river storage. If the mean runoff from GRUN is similar to GRDC observed discharge, one can infer that the poorer performance in the time series is due to exclusion of river routing.*
**Reply 7:**
Yes, as stated in the paper, we did not account for river routing when evaluating GRUN using GRDC river discharge observations. Disagreement in the time series is only visible for very large river basins where routing should be taken into account (e.g. Amazon) or in basins where water abstractions are very high (i.e. Nile).

**Comment 8:**
*Is it possible that the poorer performance in the humid basin is related to lack of data values similar to those in Amazon and Congo in the training dataset?*
**Reply 8:**
We do not observe a poor performance in humid basin. Actually, GRUN shows slightly lower performance in arid regions. We recognize that additional streamflow data in these regions would likely increase the accuracy of GRUN, which is one of the reasons underlying our concluding remark of the paper that calls for increased mobilization of discharge data around the globe.

**Comment 9:**
*In the comparison of global volumes, the GRUN also falls in the lower range. I am curious if this underprediction is coming from the lower values in the tropics which dominate the contribution to global runoff. This can be checked by having a grid-to-grid comparison (something like hexbins in Figure 3) against previous products or model simulations*
**Reply 9:**
In figure 8c we provide the spatial difference between the long-term runoff of GRUN and the multi-model ensemble mean (MMM) of ISIMIP2a hydrological simulations. We identified a hotspot region in Bangladesh were GRUN estimates are consistently lower than the hydrological model simulations. This deviation is also visible in Fig. 8b in the latitudinal band between 20 and 30° N. GRUN falls in the lower range of ISIMIP2a especially because of this region and a tendency for lower runoff rates (compared to MMM) in the subtropics (see Figure 8b). In the tropics, the GRUN reconstruction compares similarly to MMM.

**Comment 10:**
*If the underprediction is clear, it makes me wonder if predicting the runoff ratio (runoff to precipitation) would be beneficial.*
**Reply 10:**
The scope of study is the reconstruction of monthly runoff rates, which are closely related to the observable streamflow, using an already established method, which is in the spirit of ESSD. Focusing on the runoff ratio instead would imply that new methods have to be developed which goes beyond the scope of ESSD. Still, we thank the reviewer for this suggestion which would be worth investigating in an independent piece of research.

**Comment 11:**
*The GRDC discharge observations are likely to be affected by anthropogenic uses, as the manuscript rightly points out several times. Therefore, wouldn't it make sense to evaluate the GRUN product against previous mean total runoff from Beck et al., 2015 which uses fewer observation stations but more predictors (also based on neural network)?*

**Reply 11:**
In figure 3c we report the agreement between observed and predicted long-term mean runoff of 5544 grid cells. In this study, R2 is slightly higher (0.92, Fig. 3) than in Beck et al., 2015 (0.88, Table 5).
We carefully thought to compare QMEAN from Beck et al., 2015 against GRUN. However, QMEAN in Beck et al., 2015 is obtained by establishing a regression between QMEAN (estimated for every catchment over different time periods and number of timesteps) and (time invariant) multiple predictors. Therefore, QMEAN in Beck et al., 2015 is not representative of a specific time period and thus we opted to avoid such comparison.

**Comment 12:**
*Wouldn't it make sense to evaluate the GRUN product against mean ET from, say, LandFlux-EVAL, part of which is coming from satellite observations?*
**Reply 12:**
In this paper, we evaluated GRUN against observed runoff and streamflow measurements. Generally, we expect discharge observations to be more accurate than global-scale ET estimators, which is the reason why we did not compare mean runoff against mean ET. Note also, that a comparison similar to that suggested by the reviewer was conducted for the European case by Gudmundsson and Seneviratne (2015)(see Fig. 11). This analysis showed that data-driven runoff reconstructions are generally consistent with ET estimates, but also highlighted the large uncertainties associated with global ET products.

**Reference**

Adam, J. C., Clark, E. A., Lettenmaier, D. P. and Wood, E. F.: Correction of global precipitation products for orographic effects, J. Clim., 19(1), 15–38, doi:10.1175/JCLI3604.1, 2006.

Beck, H. E., de Roo, A. and van Dijk, A. I. J. M.: Global Maps of Streamflow Characteristics Based on Observations from Several Thousand Catchments, J. Hydrometeorol., 16(4), 1478–1501, doi:10.1175/jhm-d-14-0155.1, 2015.

Gudmundsson, L. and Seneviratne, S. I.: Towards observation based gridded runoff estimates for Europe, Hydrol. Earth Syst. Sci., 19(6), 2859–2879, doi:10.5194/hess-19-2859-2015, 2015.

Gudmundsson, L., Wagener, T., Tallaksen, L. M. and Engeland, K.: Evaluation of nine large-scale hydrological models with respect to the seasonal runoff climatology in Europe, Water Resour. Res., 48(11), 1–20, doi:10.1029/2011WR010911, 2012.

Humphrey, V. and Gudmundsson, L.: GRACE-REC: a reconstruction of climate-driven water storage changes over the last century, Earth Syst. Sci. Data Discuss., (February), 1–41, doi:10.5194/essd-2019-25, 2019.

---

## Author Comment (AC2) · 18 Jul 2019

We would like to sincerely thank the anonymous reviewer for the detailed and positive evaluation of our manuscript. The suggestions made by the reviewer will clearly help to improve the manuscript.
In the following, we provide point-by-point answers to the reviewer's comments. For the sake of clarity, we first repeat the reviewer's comments (in italic) and then provide our response.

**Comment 1:**
*The authors chose to develop the runoff model from precipitation and temperature, while GS15 found for Europe that this simple model is outperformed by one considering full atmospheric forcing - which is available through GSWP3. This choice must be justified in the methods section. Further, GS15 showed that including land surface parameters (soil texture and slope) does not improve model performance. I can only assume that this is the reason why land surface parameters were not included in this study. This choice needs to be clearly mentioned in the manuscript and it should at least be discussed whether this assumption is also valid on global scale.*
**Reply 1:**
We would like to thank the referee for raising the question regarding the optimal set of predictor variables used as input for the machine learning based global runoff estimator. As pointed out by the referee this choice is motivated by the findings of GS15 who showed that including additional atmospheric drivers and land-parameters resulted only in marginal gains in model performance.
Generally speaking, adding additional surface parameters as predictors (e.g. soil texture and slope) might improve the model skill. However, the search for an optimal extended predictors set would go clearly beyond the guidelines of ESSD, which emphasize the need to focus on the production and presentation of novel datasets of global relevance using already established methods (and not to present methodological or process oriented research). Instead, the scope of the presented paper is to apply a setup that was already successfully to a new data source at the global scale. We therefore refrained from conducting an extensive search for additional relevant predictor variables, although this would admittedly be an interesting topic for an independent research project. Finally, the reasonable performance of the GRUN model indicates that the focus on precipitation and temperature is justifiable for this global scale analysis.
The argument for limiting the predictors to P and T will be made clear in the revised manuscript in Section 4.1 ("Model Setup").

**Comment 2:**
*GRUN runoff estimates are compared to those from an ensemble of global hydrological models. The ensemble actually consists of global hydrology and water resources models, land surface models and dynamic vegetation models. Neither type was specifically developed or is trained/calibrated to accurately predict runoff at grid cell level - and this is without question where the strength of GRUN lies. This needs to be discussed in a revised manuscript.*
**Reply 2:**
Thank you for pointing out this fact. We will discuss this point in Section 5.3 ("Benchmarking against global hydrological models") of the revised manuscript.

**Comment 3:**
*GHMs have mainly been validated against discharge observations from catchments >10000 km2, acknowledging the uncertainty in runoff estimates at grid cell level originating from model structure and parameters, climate input, and land surface properties. To make for a fair comparison, I challenge the authors to repeat the validation experiment in section 5.3 for the GRDC discharge records.*
**Reply 3:**
Thank you for this comment. We will include this comparison for all ISIMIP experiments that are driven by GSWP3 atmospheric forcing in the supporting information of the revised manuscript. The overall results of the evaluation did not qualitatively change.

**Comment 4:**

*The authors argue that GRUN estimates do not include human interference on runoff generation, thus that differences between the GRUN reconstruction and in-situ observations could potentially be used to identify streamflow stations which have a hydrological regime very different from the naturalized flow (P14, L9-10). I strongly disagree given that the screening procedures used are capable of identifying break points in the discharges records caused by the installment of (large) water infrastructure, e.g. dams and diversions, while they likely miss more gradual changes. Consequently, the discharge time series used for training the model are a blend of near natural catchments and those impacted by human activity, and it is unclear to which degree human impact is actually implicitly represented in the reconstruction. This needs to be clearly stated and discussed in the revised manuscript.*

**Reply 4:**

Thank you for this comment. We agree that the change point detection methods are mostly sensitive to the installment of large infrastructure of water abstraction/storage and not to e.g. the impact of gradual land use change. We therefore agree that GRUN is likely not entirely free from effects of human activity as mentioned by the reviewer. It is, however, also important to remember that the temporal variability of GRUN is exclusively driven by the considered atmospheric forcing data. It is therefore our evaluation that GRUN is relatively close to near-natural conditions and clearly differs from flow conditions that are heavily impacted by human activities (see e.g. Fig. 5 and associated discussion).

We will discuss all these limitations in more detail in the revised Section 5.5 ("Limitations of GRUN").

**Comment 5:**

*The extrapolative power of GRUN is somewhat overstated (in particular P14 L7-16). The developed RF model is (in the best case) capable of predicting short-term changes in monthly runoff/water availability as a function of (changes in) precipitation and temperature with all other boundary conditions constant. Changes in water availability originate from a variety of drivers, e.g. water abstraction/diversion, land use/cover change, reservoir management, and in many cases climate may not be the most important one.*

**Reply 5:**

Thank you for pointing out that some of the discussion was formulated a bit optimistic. We will relax the wording in the revised document, also putting more emphasis and highlighting caveats and limitations of the GRUN data product.

**Comment 6:**

*The authors evaluate the performance of GRUN in a cross-validation exercise. The corresponding maps displaying six performance metrics show clear regional differences in model performance, while numbers are only provided on global level (Table 1). In order to provide a more detailed picture of model performance for potential users of the data-set, I recommend to report numbers (Q5,Q50,Q95) at sub-global level, at least for the CV-SPACE experiment. This could be climate regions or the SREX regions already used by the authors.*

**Reply 6:**

Thank you for this suggestion. In the revised supplementary material, we will provide two tables summarizing the skill distribution (Q25, Q50, Q75) for SREX regions and Koppen Geiger climate zones based on the results of the CV-SPACE experiment. We will also provide boxplots showing the skill distribution for various Koppen Geiger climate zone and SREX regions (Figures S4 and S5).

**Comment 7:**

Minor and technical comments …

**Reply 7:**

Thank you very much for identifying all these issues. We corrected all these points in the revised version of the paper. The resolution of the maps has been increased and the graphics will be provided as vector format for the final production of the paper.

---

## Author Response (AR1)

**GRUN: An observations-based global gridded runoff dataset from 1902 to 2014**

Gionata Ghiggi[1,3], Vincent Humphrey[1,2], Sonia I. Seneviratne[1], Lukas Gudmundsson[1]

[1] Institute for Atmospheric and Climate Science, ETH Zurich, Universitaetstrasse 16, 8092 Zurich, Switzerland
5 [2] Division of Geological and Planetary Sciences, California Institute of Technology, Pasadena, CA, USA
[3] Environmental Remote Sensing Laboratory (LTE), EPFL, 1005 Lausanne, Switzerland

*Correspondence to*: Gionata Ghiggi (gionata.ghiggi@gmail.com)

**Abstract**

Freshwater resources are of high societal relevance and understanding their past variability is vital to water management in the
10 context of on-going climate change. This study introduces a global gridded monthly reconstruction of runoff covering the period from 1902 to 2014. In-situ streamflow observations are used to train a machine learning algorithm that predicts monthly runoff rates based on antecedent precipitation and temperature from an atmospheric reanalysis. The accuracy of this reconstruction is assessed with cross-validation and compared with an independent set of discharge observations for large river basins. The presented dataset agrees on average better with the streamflow observations than an
15 ensemble of 13 state-of-the art global hydrological model runoff simulations. We estimate a global long-term mean runoff of 38'452 km$^3$ yr$^{-1}$ in agreement with previous assessments. The temporal coverage of the reconstruction offers an unprecedented view on large-scale features of runoff variability also in regions with limited data coverage, making it an ideal candidate for large-scale hydro-climatic process studies, water resources assessments and for evaluating and refining existing hydrological models. The paper closes with example applications fostering the understanding of global freshwater dynamics,
20 interannual variability, drought propagation and the response of runoff to atmospheric teleconnections. The GRUN dataset is available  at https://doi.org/10.6084/m9.figshare.9228176  (Ghiggi et al., 2019). (NOW TEMPORARY AT https://figshare.com/s/db241b4e0baf4fdb8430 BEFORE FINAL PUBLICATION).

**1 Introduction**

[revised manuscript text omitted]

**2.2.2 Global hydrological models' simulations**

 The Inter-Sectoral Impact Model Intercomparison Project (ISIMIP) offers a framework to compare simulations and to quantify the uncertainty across hydrological and land surface models forced with equal inputs (Warszawski et al., 2014). The accuracy of GRUN is benchmarked against runoff
20  simulations for the period 1971-2010 from an ensemble of state-of-the-art global hydrological models (GHMs) participating in the second phase of ISIMIP2a Water (Gosling et al., 2017). The  GHMs simulations used in the main text are driven with the GSWP3 forcing and do not account for human impacts on river flow ("nosoc" scenario). In the supplementary material, we also provide the results based on simulations that account for direct human impacts (i.e. the "pressoc" and "varsoc" scenarios from ISIMIP2a). Further detail on the ISIMIP2a simulation setup can be found at
25  https://www.isimip.org/protocol/#isimip2a.

**3 Data selection and pre-processing**

**3.1 GSIM time series selection and pre-processing**

**Step 1. Sub-setting GSIM stations and conversion of flow volumes to runoff rates**

Runoff is defined here as all the water draining from a small land area. Runoff cannot be observed directly, but at monthly time scale, average catchment runoff can be assumed to equal the streamflow measured at the outlet divided by the catchment area, provided storage of river water (e.g. in dams, reservoirs) and/or river water losses (through e.g. channel and lake evaporation, irrigation) are minimal. Thus, runoff rates (mm/month) are obtained by dividing the GSIM river discharge ($m^3$/month) with the station's upstream catchment area ($km^2$). We then select catchments with an area comparable to the grid-cell size of the atmospheric forcing data, in order to derive observational estimates of the runoff response to changes in atmospheric forcing.

To retrieve accurate estimates of grid-cell runoff, only GSIM stations fulfilling the following criteria have been selected for further analysis:

1.  The time series has observations within the period 1902-2014 (when GSWP3 forcing is available).
2.  The original data provider reports an estimate of the drainage area. This choice is made to have the possibility to verify the geographic location of the station as well as to assess the reliability of the automated delineation of the drainage area using a digital elevation model as provided in GSIM (Do et al., 2018).
3.  GSIM provides the shape of the drainage area and the quality of the catchment delineation is flagged as "*medium*" or "*high*". This criterion imposes that the difference between the drainage area reported by the data provider and the one estimated by GSIM differ by less than 10 % (Do et al., 2018).
4.  The drainage area is between 10 and 2500 $km^2$. Very small catchments ($< 10$ $km^2$) are discarded because the uncertainty in the drainage area can significantly affect the magnitude of the runoff rates. On the other hand, catchments larger than 2500 $km^2$ are removed because their drainage area spans too many grid-cells of the atmospheric forcing.

Based on these criteria, 10042 GSIM stations are selected for further analysis.

**Step 2. Correction for mislabeled missing values**.

Manual investigation of monthly river discharge time series revealed the occurrence of multiple consecutive months with streamflow volumes exactly equal to 0 $m^3$/month, in disagreement with the observed regional runoff pattern. These artefacts likely stem from a misleading treatment of missing values (e.g. due to damaged sensors). To identify such likely missing values, all time series are screened for the presence of more than three consecutive months with values of zero. If this pattern occurs, all zero values in the monthly time series are set to "missing".

**Step 3. Remove time series with unrealistic runoff rates and short temporal coverage**.

The following criteria have been adopted to remove observations that are too sparse or physically very unlikely:

1.  Remove time series with only missing values
2.  Remove time series with negative monthly runoff rates
3.  Remove time series with less than two years of observations

4. Remove time series with monthly runoff rates higher than 2000 mm/month

This screening step gives a selection of 8211 stations.

**Step 4. Homogeneity testing**

River discharge time series can show temporal changes in the hydrological behaviour because of changing instrumentation, recalibration of streamflow rating curves, flow regulation (i.e. dam construction) and other human activities (i.e. irrigation). Automated identification of such breakpoints is usually done using statistical tests (Gudmundsson et al., 2018b). GSIM used a general-purpose procedure that was applied to all indices/time scales. In this study, the following two target-oriented change point detection methods are applied after log-transformation of the time series:

1. Univariate normal change point in mean (Chen and Gupta, 2012);

2. Univariate normal change point in variance (Chen and Gupta, 2012);

3. Univariate normal change point in normal mean and variance (Chen and Gupta, 2012).

Runoff time series are discarded when any of these tests detected a change point.

Figure 1 shows three river flow time series with different types of detected change points. Figure 1a illustrates the ability of the tests to identify gradual changes in low flow regulation or low flows measurement precision. Figure 1b displays the detection of sudden changes in the mean of the time series, e.g. caused by dam construction, river diversion or measurement errors, while Fig. 1c shows the potential in spotting subtle changes in river discharge variability possibly induced by reservoir operations.

The homogeneity testing procedure resulted in a final selection of 7264 stations .

The file "_training_stations.csv" provided in the supplementary material lists this subset of GSIM stations, while Fig. S1 shows the catchment area distribution of these stations.

**3.2 Retrieving runoff rates at the grid  -cell scale of atmospheric forcing data**

To give equal importance to high-latitude and tropical observations, the entire modelling procedure is conducted on cylindrical equal area (CEA) grid composed of cells with an area of 2500 km$^2$ and a spatial resolution of approximately 50 km. The final gridded runoff product is however projected back onto the WGS84 grid of the atmospheric forcing data.

Because of the high density of stations in some regions and the typically elongated shape of the drainage area, many runoff observations span multiple cells of the CEA grid. Thus, an observational runoff time series representative of each cell is retrieved as follow:

1. Project the GSIM catchment shape to CEA

2. For each grid-cell:

    a. Select those catchments of which the drainage area intersects the grid-cell.

    b. At each time step, take the median runoff rate of the selected catchments.

Besides reducing the over-sampling in high station density regions, this step also smooths out some sub-grid variability. Additionally, it also can reduce the effect of potential outliers (i.e. stations that have exceptionally high or low runoff rates compared to their neighbors). To avoid inhomogeneities arising from the concatenation of different runoff time series, the observational runoff time series at each grid-cell is submitted to another homogeneity testing run (as described in Sect. 3.1, Step 4).

The procedure resulted in 5094 grid-cells usable for model training, covering 8.5% of the total land area and yielding 2'703'902 monthly runoff rate observations. Hereinafter, the grid-cell runoff time series are referred to as the runoff observations and Fig. 2 shows their spatio-temporal coverage.

**3.3 Selection and pre-processing of GRDC time series**

To obtain an independent dataset for assessing the accuracy of GRUN in large river basins, streamflow stations with catchment area larger than 10'000 km$^2$ are selected from the GRDC reference dataset. Although most of these stations are included in the GSIM collection, they are not used for model training because only catchments with area smaller than 2500 km$^2$ are used to derive grid-cell runoff observations (Sect. 3.1, Step 1).

The GRDC time series are subject to the pre-processing steps 1 to 4 detailed in Sect. 3.1 to discard streamflow records of low quality. This procedure results in a selection of 379 large river basins.

**4 Methods**

**4.1 Model Setup**

For the first time, GS15 and GS16 have used a ML algorithm to estimate monthly runoff at continental-scale and Ghiggi (2018) explored the utility of a wide range of algorithms to improve the task. Based on these findings, the present study employs the Random Forest (RF) algorithm (Breiman, 2001). RF is a ML algorithm which averages a set of randomized regression trees (Breiman et al., 1984) trained on different subsets of the original data. A regression tree divides  the predictor space into high-dimensional rectangles by means of recursive binary splits. The predicted value of a new observation is the average of the observations used in the training process located in the region of the predictor space to which the new predictor values belongs. By averaging the predictions of several randomized regression trees built on different training data, RF improves the final accuracy of the runoff estimates.

The monthly runoff rate (R) is modelled as a function of monthly precipitation (*P*) and monthly near surface temperature (*T*) as

$$R_{s,t} \; = \; f\left(\tau(P_{s,t}), \tau(T_{s,t})\right) \; (1)$$

where:

- f corresponds to the RF model (RFM);

- *s* represents the identifier of the CEA gridcell;

- *t* is the timestep;

- *τ* is a time lag operator that provides information about meteorological conditions of the past six months to allow the RFM to approximate water storage effects that influence the runoff generation process. This differs from GS15 and GS16 which used a time lag operator of 12 months. The reasons behind this change are a reduction in training time of RFM and to decrease collinearity between predictors (caused by the seasonal cycle).

Both precipitation and runoff observations are log-transformed before model training to adjust the skewed distribution of the data and avoid that only a small number of high flow events dominate the optimization of the squared error loss. Once the RFM is trained, gridded precipitation and temperature data are fed to the model to obtain the final runoff reconstruction. Finally, the log-transformation of the predicted runoff values is inverted to derive runoff rates in conventional units.

The decision to only consider precipitation and temperature as explanatory variables, is motivated by GS15 who found that the inclusion of other atmospheric variables as well as selected land parameters (topography and soil texture) did not significantly improve the overall accuracy of the estimate. Furthermore, reducing the number of predictor variables also helped to reduce computational costs significantly. While a more extensive screening of other land parameters is beyond the scope of this study, this could be the subject of potential future research.

[revised manuscript text omitted]
. Figures S5 and S6 show the distribution of CV-SPACE skills of relBIAS, NSE and $R2_{clim}$ for each Koppen Geiger (KG) climate zone (Peel et al., 2007) as well as the various SREX regions. The $25^{th}$, $50^{th}$, and $75^{th}$ quantiles of these skills distribution are reported in the "KG_CV_SPACE_Skills.csv" and "SREX_CV_SPACE_Skills.csv" files provided in the supplementary material. Figures S7 and S8 show the cumulative distribution function of the monthly runoff rates and the monthly standardized anomalies for

10  different climate zones respectively. In dry climates (group B of KG) we note an overestimation of GRUN in the lower part of the runoff rates distribution compared to the observation, although in terms of standardized anomalies, the cumulative distribution of the of the estimate agrees very well with the observations.

**5.2 Basin-scale validation**

Figure 5a illustratesevaluates the accuracy of GRUN using the selection ofselected GRDC reference streamflow stations, (Sect.

15  3.3), while Fig. 5b shows the observational agreement of river flow timeseriestime series for some selected basins displayed in Fig. 5a.  The temporal evolution of river flow is in general well captured and an underestimation of the peak flow volume is only evident for the Mekong river. For the Ebro the agreement between observations and GRUN startstarts to decrease from 1965 ahead. The dynamics are no more well captured, and GRUN estimates are constantly higher than the GRDC observations. These discrepancies might be caused by the intensive irrigation and reservoir activities which have altered the natural

20  hydrological regime of the basin. In that respect, it is interesting to notice that the NSE spatial pattern in Fig. 5a showshows many similitudes with the estimated amount of runoff stored by engineered impoundments reported in Vörösmarty et al. (2004)(2004): low NSE scores tends to correspond to higher fractions of water impoundment. Both the Nile and Colorado river basins are an exceptional example of the human-induced river flow alterations.

However, human activities are not the only cause of discrepancies between GRUN-based river discharge estimates and the

25  observations. In the Amazon river, the negative NSE value and the visible phase lag between the estimated and the observed time series ismight not be caused by an inaccurate runoff reconstruction, but rather related to the fact that river discharge is simply estimated using the average runoff within the basin. without taking water travel times into account. Indeed, for such a very large river basin, a routing model accounting for water travel times would be necessary to correctly reproduce the river flow dynamics also at monthly timescales. Figure S8 shows the spatial distribution of the remaining skill scores (e.g. other

30  than NSE) for the GRDC basins.

**5.3 Benchmarking against global hydrological models**

In this section, we benchmarked the performance of GRUN against well-established GHMs at two different scales.

Figure 6 compares the distribution of the skill scores for the CV-SREX and CV-SPACE experiments against the skill of the ISIMIP2a GHMs runoff simulations

from the "nosoc" experiment at the grid-cell scale. CV-SPACE always has higher skills than CV-SREX and outperforms all ISIMIP2a GHMs runoff simulations and their multi-model ensemble mean (MMM) except for *relBIAS* and

*rSD.* Overall, the GRUN cross-validation experiments show a tendency to underestimate runoff although the skill spread of *relBIAS* is reduced compared to the ISIMIP2a models. Among the GHMs there is not a clear tendency to under/overestimate runoff. The same applies for the variability (*rSD*). The dynamics of runoff (*R2*) are  better reproduced by GRUN than the considered GHMs, and the overall *NSE* skill score distribution is  better for GRUN than for the ISIMIP2a GHMs simulations. The anomalies (*R2$_{anom}$*) are also  better reproduced by GRUN, and CV-SREX outperforms all the single GHMs.

Finally, *R2$_{clim}$* demonstrates that GRUN reproduces much better the seasonal cycle than the GHMs. Previous studies already showed that GHMs struggle in reproducing the seasonality of runoff (Gudmundsson et al., 2012; Gudmundsson and Seneviratne, 2015). Similar conclusions can be drawn when benchmarking GRUN against the "pressoc" and "varsoc" experiments from the ISIMIP2a runoff simulations (Fig. S2 and S3 respectively).

Because GHMs are typically not calibrated at the grid-cell scale (unlike GRUN), we also benchmark GRUN against ISIMIP2a GHMs simulations in large river basins using the selection of GRDC reference stations with catchment area larger than 10'000 km$^2$ detailed in Sect. 3.3. The results for the ISIMIP2a "nosoc","pressoc" and "varsoc" scenarios are reported in Fig. S9, S10 and S11 respectively. The dynamics of runoff (*R2*), the anomalies and the climatology (*R2$_{clim}$*) are still better reproduced by GRUN than the ISIMIP2a GHMs across all scenarios. The average *relBIAS* of GRUN is close to 0 while the variability is slightly overestimated: this contrasts the results obtained at the grid-cell scale where GRUN tends to underestimate the variability (*rSD*) compared to the observations. Figure S12 also provides a comparison of simulated river discharge from ISIMIP2a against 50 GRUN realizations (see Sect. 4.2) for the same time series displayed in Fig. 5, highlighting the larger scatter of conventional GHMs, likely due to structural and parameter uncertainties.

**5.4 Sensitivity of the runoff estimates to the observations used for training**

An ensemble of 50 runoff reconstructions trained on different subsets of observations (Sect. 4.2) is used to assess the sensitivity of GRUN to the observations used for training. Figure 7 shows the long-term mean of the monthly ensemble standard deviation and coefficient of variation (defined as the standard deviation divided by the mean). Regions characterized

by higher runoff rates show higher standard deviation (Fig. 7a), but this variability across the realizations is small (< 20 %) compared to the runoff magnitude (Fig. 7b). With the exception of arid regions, the coefficient of variation is generally below 0.2 (Fig. 7b).

To put  the sensitivity of GRUN in relation to the observations used for training, Fig. 8 compares the
5  annual runoff volumes of the GRUN realizations against the state-of-the-art GHMs participating in ISIMIP2a.

The global long-term mean runoff volume estimated by GRUN (38'452 km3/year) lies within the lower range of ISIMIP2a GHMs (Fig. 8a) and generally agrees with other global terrestrial discharge estimates (Table 3). The uncertainty attributed to the selection of training observations (shaded area in Fig. 8a) of the global GRUN runoff volume is far smaller than the spread introduced by different physical representations of the hydrological processes in the GHMs. The
10  uncertainty introduced by the selection of training observations increases proportionally with the magnitude of the runoff rates and is highest in the tropics (Fig. 8b). Reversely, the spread of the GHMs tends to be constant across all latitudes. GRUN has almost always latitudinal mean runoff rates lower than the MMM and goes beyond the GHMs range only between 20° and 30° latitudes North. This pattern is mainly related to the relatively low runoff estimates in GRUN in Northeast India and Bangladesh compared to the GHMs (Fig. 8c).

15  **5.5 Limitations of GRUN**

The streamflow observations used for model training underwent careful preprocessing and screening steps to remove
20   time series presenting sudden changes in the hydrological signature. Therefore, and because the product is solely forced with precipitation and temperature, GRUN is not able to explicitly account for the effects of local human river flow regulation
25  (dam operations in particular) on the reconstructed hydrological regimes. However, we note that some streamflow observations impacted by irrigation or other land- and water management practices have likely not been removed, especially if the magnitude of water abstraction/returns did not alter the monthly hydrograph sufficiently to identify a change point or if the time series is not long enough to cover past periods of near-natural streamflow. This may be one of the reasons for the overestimation of runoff rates in several arid regions (Fig. 4a-b) known for intensive-irrigation activities (Wriedt et al., 2009; Siebert et al., 2015).
30  To some extent, the impact of past land-use changes on water availability might be implicitly accounted for in GRUN. For example if the GSWP3 bias-corrected reanalysis captured regional changes in precipitation and temperature which were induced by human activities (e.g. Davin et al., 2007; De Angelis et al., 2010; Luyssaert et al., 2014; Alter et al., 2015; Thiery et al., 2017) or if water management practices are altered gradually together with a climate change signal (e.g. irrigation may

increase with decreasing precipitation). Any changing pattern in water availability emerging from GRUN is however solely conditioned by trends of the GSWP3 forcing and the runoff observations used for model training. Thus, our evaluation is that GRUN estimates likely lie closer to near-natural runoff conditions than to human-regulated conditions (e.g. see the Nile river estimate in Fig. 5b), even though we cannot exclude that GRUN implicitly includes some human effects due to the various reasons mentioned above. Finally, we note that the accuracy of the runoff rates in mountainous regions is likely not optimal. The coarse resolution of the considered meteorological forcing does not allow capturing the sub-grid variability of precipitation and temperature that governs e.g. snow-melt volume and timing in such regions. Although the statistical model could implicitly account for homogenous biases in the forcing dataset and streamflow observations, the reader must be aware of possible inconsistent water balance in such regions. Glacier melting is also not explicitly accounted for in GRUN.

To some extent, the impact of past land use changes on water availability might be implicitly accounted for in GRUN if the GSWP3 reanalysis captured the changes in precipitation and temperature which were induce by such human activities through land atmosphere interactions. Any changing pattern in water availability emerging from GRUN is however solely conditioned by trends of the GSWP3 forcing and the runoff observations used for model training.
Finally, the accuracy of the runoff rates in mountainous regions is likely not optimal. The spatial resolution of the considered meteorological forcing does not allow to capture the sub-grid variability of precipitation and temperature that governs e.g. snow-melt volume and timing in such regions. Glacier melting is also not accounted for in GRUN.

**6. Example Applications**

**6.1 Runoff climatology**

Figure 9a displays the annual runoff climatology derived as the long-term mean of the GRUN reconstruction covering the 1902-2014 period. Figure 9a shows that longLong-term mean runoff rates can differ by three orders of magnitude across the globe. The, with highest runoff rates are observed mostly in the tropics and in large mountain ranges. The, and lowest rates in the extratropics show low runoff rates, in correspondence with theand major world deserts such as the Sahara. Monthly climatologies are provided in the supporting information (Fig. S1).

S13). Figure 9b and 9c show the months with the maximum and minimum of the long term mean seasonal cycle. In the Northern Hemisphere, regions exposed to winter snow accumulation have the lowest runoff during the coldest months in winter and a runoff peak toward the end of spring which is related to the melting of theas a result of snowpack. melting and decreasing terrestrial water storage (Humphrey et al., 2016). In the humid mid-latitudes, evapotranspiration closely follows the seasonal cycle of surface temperature, causing the lowest (highest) runoff to occur prevalently during the summer (winter) months. In the tropics, the month with maximum runoff tends to occur during the rainy season. In the Northern tropics this occurs between August and September, while in the Southern tropics between February and April, following which follows the migration of the Intertropical Convergence Zone (Schneider et al., 2014)(Schneider et al., 2014).

**6.2 Trends in reconstructed runoff**

GRUN can be used to investigate changing freshwater availability. Trends in observed (Fig. 10a) and estimated (Fig. 10b) annual runoff for the period 1971-2010 are computed using Sen's slope (Sen, 1968) and expressed in absolute and relative terms. Overall the reconstructed trends are in line with other reported findings (Gudmundsson et al., 2018a) and closely resemble the observed trends.

In Europe, the Mediterranean regions exhibits a decrease in annual runoff, while in Central and Northern Europe there is a tendency to increasing runoff rates. This pattern is in agreement with previous studies (Stahl et al., 2010, 2012) and was recently attributed to anthropogenic climate change (Gudmundsson et al., 2017). In the Eastern and Western USA negative trends occur, while large portions of the Mississippi river basin show increasing runoff.

In the tropics, the Amazon basin shows a substantial decrease in annual runoff rates and a reduction of freshwater discharge to the Atlantic Ocean has the potential to impact the Atlantic and the Northern Hemisphere climate (Vizy and Cook, 2010; Jahfer et al., 2017). In light of the human pressure to which this basin is currently exposed (Castello and Macedo, 2016; Latrubesse et al., 2017) and the uncertain impact of deforestation on river flow (D'Almeida et al., 2007; Spracklen et al., 2012; Lawrence and Vandecar, 2015; Spracklen and Garcia-Carreras, 2015), the causes and consequences of such trends should be investigated in more detail. Similarly, the drying tendency observed in many regions of the Congo Basin could affect the Eastern Equatorial Atlantic climate variability (Materia et al., 2012). Reversely, tropical area situated in Southeast Asia experiences an increase in runoff.

~~The monthly resolution of GRUN also allows to investigate these changes at subseasonal time scales (Fig. S2), which might e.g. be relevant for water resource assessments because neglecting the seasonal fluctuations can cause underestimation of water scarcity (Mekonnen and Hoekstra, 2016). In addition to changes in magnitude, the GSIM dataset offers also the possibility to analyze shifts in the seasonality of the hydrological regimes. Figure S3 provide an overview of the months in which the minimum and maximum runoff volumes occurred at the beginning and at the end of the 20th century.  Over Europe, the pattern of change is consistent to ones emerging in recent studies (Blöschl et al., 2017; Hall and Blöschl, 2018).~~

The monthly resolution of GRUN also allows investigating these changes at sub-seasonal time scales (Fig. S14), which might e.g. be relevant for water resource assessments because neglecting the seasonal fluctuations can cause underestimation of water scarcity (Mekonnen and Hoekstra, 2016). In addition to changes in magnitude, the GSIM dataset offers also the possibility to analyze shifts in the seasonality of the hydrological regimes. Figure S15 provide an overview of the months in which the minimum and maximum runoff volumes occurred at the beginning and at the end of the 20th century. Over Europe for example, Fig. S15 shows evidence for earlier occurrence of maximum runoff, which is consistent with changes in snowmelt timing already reported 
[revised manuscript text omitted]
; (v) to reconstruct droughts in the last-millennium in combination with tree rings (Nicault et al., 2008; Cook et al., 2010a, 2010b; Meko et al., 2012; Cook et al., 2015); (vi) to benchmark regional streamflow archives and hydrological reconstructions (Wang et al., 2009; Wu et al., 2011; Caillouet et al., 2017; Mishra et al., 2018; Moravec et al., 2019; Smith et al., 2019); and (vii) to address other scientific challenges in water cycle research (Wagener et al., 2010; Montanari et al., 2013; Greve et al., 2014; Trenberth and Asrar, 2014; Hegerl et al., 2015).

We conclude by remarking that this dataset would not have been possible without the mobilization of national and international hydrological archives. This study shows the benefit of a wider access to hydrological data collected by various institutions worldwide. We call for a continuation of the international efforts to reduce political and technical barriers for the exchange of hydrometeorological data across the scientific community.

**8.   Data availability**

The GRUN dataset based on GSWP3 forcing is publicly available in NetCDF-4 format (Ghiggi et al., 2019) and can be freely downloaded at https://doi.org/10.6084/m9.figshare.9228176.

**9.   Competing interests**

The authors declare that they have no conflict of interest.

**10.  Author contribution**

LG initiated this investigation. GG, VH, SIS and LG designed the study. GG developed the model code and performed the analysis. GG prepared the manuscript with contributions from all co-authors.

**11.  Acknowledgments**

We thank Prof. Dr. Hyungjun Kim for providing us with early access to the GSWP3 dataset and GRDC for the river discharge observations. SIS acknowledges partial support from the ERC DROUGHT-HEAT project funded by the European Community's Seventh Framework Programme (grant agreement FP7-IDEAS-ERC-617518). VH acknowledges support from the Swiss National Science Foundation (P400P2_180784).

[revised manuscript text omitted]

**Figure 11t.** Interannual variability of runoff and its relation to modes of climate variability. a) Hovmöller diagram of standard runoff anomalies (reference period 1902-2014). Vertical dashed lines indicate onset of El Nino events. b) Time series of the Multivariate ENSO Index (MEI). Red and blue shades characterize the intensity of El Nino and La Nina conditions respectively. c) Correlation of the MEI with monthly runoff anomalies. d) Relationship of European runoff anomalies with the North Atlantic Oscillation (NAO).

[Figure]

[Figure]

**Figure 12.** Three extreme drought events as reconstructed by GRUN. a)  European summer drought in 1976. b) U.S. Dust Bowl in 1934. c) Ethiopian famine in 1984.

**Table 1.** Median values of skill scores for CV-SREX and CV-SPACE

|  | CV-SREX | CV-SPACE |
|---|---|---|
| relBIAS | -0.123 | -0.109 |
| rSD | 0.794 | 0.814 |
| R2 | 0.594 | 0.734 |
| NSE | 0.394 | 0.610 |
| $R2_{anom}$ | 0.350 | 0.522 |
| $R2_{clim}$ | 0.874 | 0.946 |

**Table 2.** Median values of skill scores computed for the large GRDC river basins

| relBIAS | rSD | R2 | NSE | $R2_{anom}$ | $R2_{clim}$ |
|---|---|---|---|---|---|
| 0.047 | 1.004 | 0.738 | 0.525 | 0.394 | 0.916 |

**Table 3.** Comparing global long-term mean runoff from GRUN against values reported in the literature. GRUN estimates are obtained by considering the same time span and spatial coverage of the reported studies. Values in parenthesis denote the uncertainty range reported in some studies.

| Reference | Runoff ($km^3$ $yr^{-1}$) | | Time period | Greenland | Antarctica |
|---|---|---|---|---|---|
|  | Reference | GRUN[b] |  |  |  |
| Dai and Trenberth, 2002[a] | 37288 | - | - | No | No |
| Fekete et al., 2002 | 39319 | - | "Climatology" | No | No |
| Döll et al., 2003 | 36687 | 39173 | 1961-1990 | Yes | No |
| Syed et al., 2009[a] | 30354 | 36565 | 2003-2005 | No | No |
| Wisser et al., 2010 | 37984 | 38833[c] | 1901-2002 | Yes | No |
| WaterMIP (Haddeland et al., 2011) | 42000-66000[d] | 37994 | 1985-1999 | No | No |
| Clark et al., 2015[a] | 44200 | 38942 | 1950-2008 | Yes | Yes |
| Rodell et al., 2015[a] | 45900 | 38360 | 2000-2010 | Yes | Yes |
| Müller Schmied et al., 2016 | 41298 (39200-42200) | 38628 | 1971-2001 | No | No |
| eartH2Observe (Schellekens et al., 2017) | 46268 (38652-55877) | 38163 | 1979-2012 | No | No |
| ISIMIP2a nosoc scenario (GSWP3) | 45180 (35997-57323) | 38452 | 1971-2010 | No | No |

[a]The long-term mean is obtained by extrapolation from continental-scale river-discharge observations or water balance.
[b]Antarctica is never included in the GRUN estimate of the global long-term mean runoff. Greenland is considered only if included in the reference dataset.

[c]GRUN long-term mean runoff is computed for the period 1902-2002.
[d]Haddeland et al. (2011) report that the CRU land mask used to rescale global mean runoff (excluding Antarctica and Greenland) has an area of $1.44 \cdot 10^8$ km$^2$, while the correct area value should range around $1.33 \cdot 10^8$ km$^2$.

5     (Clark et al., 2015; Dai and Trenberth, 2002; Döll et al., 2003; Fekete et al., 2002; Haddeland et al., 2011; Müller Schmied et al., 2016; Rodell et al., 2015; Schellekens et al., 2017; Syed et al., 2009)

**Supplementary Material**

[Figure]

**Supplementary Figure 1:** Runoff monthly climatology for the period 1971-2010.

[Figure]

Relative Runoff Trend [%/yr]

**Supplementary Figure 2:** Runoff monthly trends for the period 1971-2010.

[Figure]

**Supplementary Figure 3:** Changes in runoff timing. a) Minimum runoff b) Maximum runoff.

(Clark et al., 2015; Dai and Trenberth, 2002; Döll et al., 2003; Fekete et al., 2002; Haddeland et al., 2011; Müller Schmied et al., 2016; Rodell et al., 2015; Schellekens et al., 2017; Syed et al., 2009).